# Benchmark performance of low-cost Sb$_2$Se$_3$ photocathodes for unassisted solar overall water splitting

Wooseok Yang [1,4], Jin Hyun Kim[2,4], Oliver S. Hutter [3], Laurie J. Phillips [3], Jeiwan Tan[1], Jaemin Park[1], Hyungsoo Lee[1], Jonathan D. Major[3 ✉], Jae Sung Lee[2 ✉] & Jooho Moon [1 ✉]

Determining cost-effective semiconductors exhibiting desirable properties for commercial photoelectrochemical water splitting remains a challenge. Herein, we report a Sb$_2$Se$_3$ semiconductor that satisfies most requirements for an ideal high-performance photoelectrode, including a small band gap and favourable cost, optoelectronic properties, processability, and photocorrosion stability. Strong anisotropy, a major issue for Sb$_2$Se$_3$, is resolved by suppressing growth kinetics via close space sublimation to obtain high-quality compact thin films with favourable crystallographic orientation. The Sb$_2$Se$_3$ photocathode exhibits a high photocurrent density of almost 30 mA cm$^{-2}$ at 0 V against the reversible hydrogen electrode, the highest value so far. We demonstrate unassisted solar overall water splitting by combining the optimised Sb$_2$Se$_3$ photocathode with a BiVO$_4$ photoanode, achieving a solar-to-hydrogen efficiency of 1.5% with stability over 10 h under simulated 1 sun conditions employing a broad range of solar fluxes. Low-cost Sb$_2$Se$_3$ can thus be an attractive breakthrough material for commercial solar fuel production.

[1] Department of Materials Science and Engineering, Yonsei University, 50 Yonsei-ro, Seodaemun-gu, Seoul 03722, Republic of Korea. [2] School of Energy and Chemical Engineering, Ulsan National Institute of Science and Technology, Ulsan 44919, South Korea. [3] Stephenson Institute for Renewable Energy, Physics Department, University of Liverpool, Liverpool L69 7XF, UK. [4] These authors contributed equally: Wooseok Yang, Jin Hyun Kim. ✉email: Jon.Major@liverpool.ac.uk; jlee1234@unist.ac.kr; jmoon@yonsei.ac.kr

Molecular hydrogen production via photoelectrochemical (PEC) splitting of water is a promising solution toward a zero-carbon-based society. In order to produce sustainable H$_2$ for a scale commensurate with the global energy demand, high efficiency and low cost, both of which highly depend on the semiconductor materials used in the PEC system, should be achieved. As the solar-to-hydrogen (STH) efficiency of PEC devices using expensive photovoltaic-grade III–V semiconductors has approached the theoretical maximum[1], the research community is now recognising the importance of exploring low-cost materials exhibiting good optoelectronic properties[2]. With this consideration, nearly all recent critical evaluations[3–5] pertaining to artificial photosynthesis have urged the development of new light absorbers. According to theoretical calculations[6,7], two different light absorbers—having a band gap ($E_g$) of ~1.8 eV for the top electrodes and 1.0–1.3 eV for the bottom electrodes—are required for realising STH efficiencies of >23% in the D4 (dual light absorber: four photons to one H$_2$) tandem cell. Further requirements for suitable semiconductors for PEC water splitting include a large light absorption coefficient ($\alpha$), high mobility, easy processability without secondary phases, and stability to photocorrosion. In this regard, most emerging cost-competitive photoelectrode materials for PEC water splitting, such as TiO$_2$ ($E_g \sim 3.2$ eV)[8], Fe$_2$O$_3$ ($E_g \sim 2.2$ eV)[9], BiVO$_4$ ($E_g \sim 2.4$ eV)[10], and Cu$_2$O ($E_g \sim 2.0$ eV)[11], are deemed unsuitable for the bottom photoelectrode due to their large $E_g$. Although the $E_g$ of Cu$_2$ZnSn(S,Se)$_4$ semiconductor, which is another earth-abundant material applicable for PEC water splitting, varies from 1.0 to 1.5 eV depending on the composition, the pure-phase compound suffers from severe difficulty of synthesis due to the narrow stoichiometric window[12]. It is worth mentioning that the secondary phase issue, i.e. the formation of undesirable detrimental phases during processing, could pose a major obstacle to large-scale commercialisation. In recent years, various low-cost semiconductors have emerged, such as Cu$_2$S ($E_g \sim 1.5$ eV)[13], CuFeO$_2$ ($E_g \sim 1.5$ eV)[14], CuBi$_2$O$_4$ ($E_g \sim 1.7$ eV)[15], CuSbS$_2$ ($E_g \sim 1.5$ eV)[16], and SnS ($E_g \sim 1.3$ eV)[17]. However, none of them have satisfied all the requirements for an ideal semiconductor for PEC water splitting. Therefore, a breakthrough material for realising practical STH conversion is urgently required[4].

Sb$_2$Se$_3$ offers many advantageous properties that make it a nearly ideal semiconductor material for PEC water splitting. First, it has a small $E_g$ of 1.1–1.2 eV. In addition, it is a low-cost semiconductor (the cost of Sb is similar to that of Cu)[18] and exhibits attractive optoelectronic properties ($\alpha > 10^5$ cm$^{-1}$ and high mobility of ~10 cm$^2$ V$^{-1}$ s$^{-1}$)[19,20]. Moreover, the thermodynamic properties of the Sb-Se system allow only stable orthorhombic Sb$_2$Se$_3$ phases without any secondary phases[21], avoiding severe secondary phase issues commonly observed in other multivalent compound materials. It has also been reported that Sb$_2$Se$_3$ is intrinsically stable towards photocorrosion in strong acidic media unlike Cu$_2$O, which is vulnerable to photocorroson[22]. Owing to these merits, the photocurrent density of the Sb$_2$Se$_3$ photocathode has improved rapidly from the initially reported value of 2.5 mA cm$^{-2}$ to 17.5 cm$^{-2}$ at 0 V against the reversible hydrogen electrode (RHE) over the short period of its development[23–26]. However, it is still far below the theoretical maximum (about 40 mA cm$^{-2}$, assuming a 1.2 eV $E_g$ and a 100% incident photon-to-current efficiency (IPCE)). In addition, it has been reported that the onset potential of Sb$_2$Se$_3$ photocathodes can be enhanced up to 0.47 V$_{RHE}$[27], but the photovoltage and fill factor should be further improved to realise unassisted water splitting.

As Sb$_2$Se$_3$ has a one-dimensional (1D) crystal structure, it tends to grow with the morphology of 1D nanostructures, which makes

synthesis of compact Sb$_2$Se$_3$ thin films difficult[28]. In the case wherein a light-absorbing material has a low absorption coefficient and poor electrical properties (e.g. short carrier diffusion length), 1D nanostructuring could be an attractive strategy for enhancing the performance. However, in the previously reported Sb$_2$Se$_3$ photocathodes, probably owing to its good optoelectronic properties, planar-type Sb$_2$Se$_3$ exhibited better performance compared with that of its elongated 1D structured counterpart[24,27]. One possible reason is that the incomplete coverage of Sb$_2$Se$_3$ films, due to the complicated 1D morphology, can result in direct contact between the substrate and an n-type layer, which acts as a recombination centre. In addition, because the carrier transport in Sb$_2$Se$_3$ along the [001] direction is more efficient than that along the [010] and [100] directions owing to its anisotropic crystallographic nature[20], the p–n junction should be formed along the [001] direction to maximise the separation of the photo-generated charges. In the 1D Sb$_2$Se$_3$ system, however, the p–n junction generally is formed along the [010] or [100] directions, resulting in inefficient charge separation. Thus compact film-type Sb$_2$Se$_3$ with favourable crystallographic orientation for efficient charge separation (i.e. vertically aligned (Sb$_4$Se$_6$)$_n$ nanoribbons) should be obtained to achieve high-performance Sb$_2$Se$_3$ photocathodes. However, as molecular inks for Sb$_2$Se$_3$ already contain 1D [Sb$_4$Se$_7$]$^{2-}$ chains in the solution[24,27], it is a daunting task to obtain compact thin Sb$_2$Se$_3$ films via solution processing.

Here we report high-quality dense Sb$_2$Se$_3$ thin films obtained by the close space sublimation (CSS) method, which is known to be a low-cost, large-area, high-yield deposition technique for preparing light-absorbing semiconductors, such as cadmium telluride[29] and CH$_3$NH$_3$PbI$_3$[30]. We implement a fast cooling strategy as well as the two-step CSS method[31], enabling pin-hole-free and smooth Sb$_2$Se$_3$ thin films with well-oriented (Sb$_4$Se$_6$)$_n$ ribbons. The resulting Sb$_2$Se$_3$ thin-film-based photocathodes reveal a highest photocurrent density close to 30 mA cm$^{-2}$ at 0 V$_{RHE}$. The performance of the newly developed Sb$_2$Se$_3$ photocathode is utilised by unassisted overall water splitting PEC cells after judiciously optimising the photovoltage and electrolyte compatibility to the BiVO$_4$ photoanode. The D4 tandem cell of Sb$_2$Se$_3$–BiVO$_4$ in a neutral phosphate buffer successfully demonstrates an impressive STH efficiency of 1.5% with 10 h of stability. Realisation of the Sb$_2$Se$_3$-based tandem device, capable of harvesting a broad range of photons with the readily obtained earth-abundant semiconductor, allows us to envision practical solar hydrogen production via efficient and cost-competitive PEC water splitting.

## Results

**Fabrication of compact and well-oriented Sb$_2$Se$_3$ thin films**. In order to avoid the formation of the thermodynamically favourable 1D structure as well as induce the metastable thin film morphology of Sb$_2$Se$_3$, we tried to kinetically suppress the rearrangement of (Sb$_4$Se$_6$)$_n$ nanoribbons by employing a two-step deposition where a low-temperature (340 °C) seed layer was deposited prior to the high-temperature (460 °C) deposition[31]. In general, the CSS system employs one-step deposition at a high temperature (e.g. 400–550 °C for CdTe) to achieve a suitable growth rate[29]. Owing to the highly anisotropic nature of Sb$_2$Se$_3$, however, the low-temperature deposition is an imperative step to obtain a compact morphology. At the first deposition step of 340 °C, a compact thin film could be obtained, whereas an elongated 1D structure with incomplete coverage was observed when a temperature of 460 °C was applied directly (Supplementary Fig. 1). Figure 1a–c show the microstructures of Sb$_2$Se$_3$ thin films deposited on an Au/fluorine-doped tin oxide (FTO) substrate by the two-step CSS deposition process (i.e. successive depositions at 340 and 460 °C followed by cooling the chamber

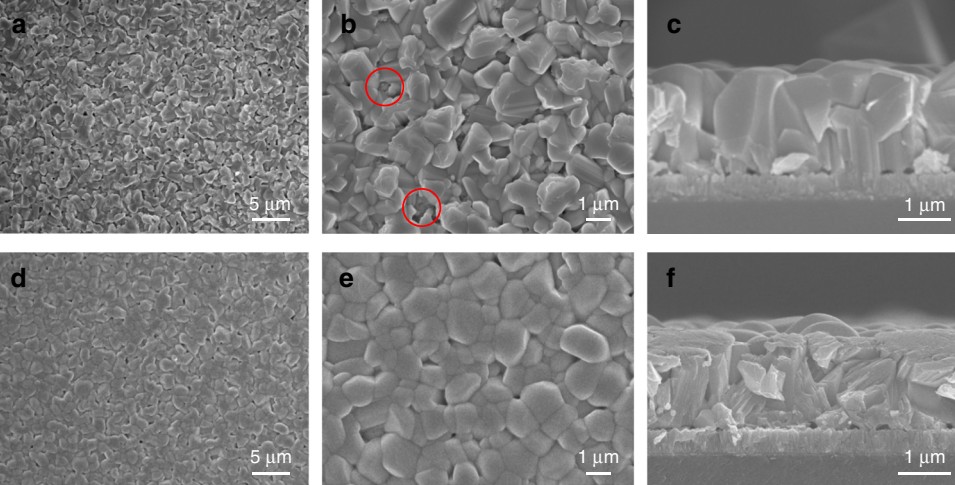

**Fig. 1 Microstructures and crystallographic orientation of Sb₂Se₃ thin films. a–f** SEM images of **a–c** slow-cooling Sb₂Se₃ and **d–f** fast-cooling Sb₂Se₃.

naturally after deposition). The role of the Au layer in Sb₂Se₃ photoelectrodes, which facilitates the transfer of photo-generated holes, has been discussed in previous reports on Sb₂Se₃ for PEC water splitting[26,32] and thin-film solar cells[33]. Despite the film-type morphology, there were some pin-holes resulting in exposure of the substrate (red circles in Fig. 1b) with the faceted morphology of the Sb₂Se₃ thin films. Interestingly, it was found that a much smoother pin-hole-free surface was achieved when N₂ gas was passed during the cooling process after deposition under the same conditions (i.e. successive depositions at 350 and 450 °C), as shown in Fig. 1d–f. We denoted the sample prepared with N₂-assisted cooling as the 'fast-cooling' sample, while the naturally cooled sample was denoted as the 'slow-cooling' sample. The cooling rates for the fast cooling and the slow cooling until it has reached 200 °C are approximately 15.7 and 11.3 °C min⁻¹, respectively. The detailed differences between the two samples, including grain size distribution and crystallographic orientations, are shown in Supplementary Figs. 2–4.

On comparing the morphological and crystallographic differences between the fast- and slow-cooling samples, it was obvious that the N₂ flow during the cooling process significantly affects the morphology and crystallographic orientation. As mentioned above, because Sb₂Se₃ has a strong 1D anisotropic nature, thermodynamically favourable sharp and facet morphologies are inevitably obtained when kinetically sufficient time is provided (i.e. slow cooling). In contrast, in case of fast cooling, which offers insufficient time to reach the thermodynamically stable morphology, a metastable smooth morphology can be achieved. This explanation is supported by the solution-processed Sb₂Se₃, in which 1D nanostructures were observed when sufficient Se precursor was provided, while the metastable planar-type Sb₂Se₃ was obtained in Se-deficient conditions[27]. In addition, a slight change in X-ray diffraction (XRD) data implied that the ribbons likely move during the cooling process, presumably due to the low melting temperature of Sb₂Se₃ (~608 °C). Thus it can be reasonably concluded that, in order to fabricate compact thin-film-type Sb₂Se₃ with favourably oriented (Sb₄Se₆)ₙ ribbons, rearrangement of the ribbons should be kinetically suppressed as much as possible to prevent the formation of thermodynamically stable 1D structures. Obtained with kinetically controlled growth, it was possible to obtain Sb₂Se₃ thin films with a compact structure as well as well-oriented ribbons, both of which are expected to be favourable for better performance.

**PEC performance of CSS-Sb₂Se₃ photocathodes.** Figure 2a, b show the PEC performance of RuO$_x$/TiO₂/Sb₂Se₃/Au/FTO photocathodes using fast- and slow-cooling Sb₂Se₃ films measured in pH 1 electrolytes. As we mentioned above, the Au layer acts as a hole-selective contact, which facilitates the transfer of photo-generated holes while blocking the electrons backflow[32,33]. Without the Au layer, Sb₂Se₃ photocathodes revealed relatively poor performance while nearly similar morphology of Sb₂Se₃ was observed (Supplementary Fig. 5), which verifies the role of the Au layer not affecting the growth of Sb₂Se₃ but assisting the transfer of photo-generated charges. The RuO$_x$ catalytic layer was deposited by the PEC method, while atomic layer deposition (ALD) was used for the TiO₂ layer, similar to a previous study[32]. In both samples, the onset potentials shifted towards a positive direction after the first scan due to activation of the RuO$_x$ catalyst[34]. The photocurrent density of the fast-cooling sample approached 30 mA cm⁻² at 0 V$_{RHE}$, which is not only the highest value obtained for a Sb₂Se₃ photocathode but also among the best observed for all photoelectrodes used in PEC water splitting so far. Note that the data shown in Fig. 2 were obtained from the best-performing device, while normally 25–30 cm⁻² at 0 V$_{RHE}$ photocurrent density was observed in the fast-cooling Sb₂Se₃-based photocathodes. The previously reported photocurrent densities of photoelectrodes for PEC water splitting are shown in Supplementary Fig. 6. Given the low-cost and relatively short history of Sb₂Se₃ as well as the simple preparation and low material usage due to the high α, the high photocurrent density of ~30 mA cm⁻² at 0 V$_{RHE}$ clearly demonstrates the strong potential of Sb₂Se₃ as a promising photocathode material.

While the fast-cooling sample revealed a record photocurrent density, the photocurrent density of the slow-cooling sample with a RuO$_x$ co-catalyst, which had a facet morphology, was relatively low (~17 mA cm⁻² at 0 V$_{RHE}$, Fig. 2b). Figure 2c exhibits the half-cell STH conversion efficiencies (HC-STH) calculated from the third scans in Fig. 2a, b according to the equation HC-STH = $I_{ph} \times (E_{RHE} - E_{H+/H2})/P_{SUN} \times 100\%$, where $I_{ph}$ is the photocurrent density obtained under an applied bias of $E_{RHE}$, $E_{H+/H2}$ is 0 V$_{RHE}$, and $P_{SUN}$ is 100 mW cm⁻². It is worth noting that the maximum value of HC-STH in the fast cooling was observed at a more positive potential of 0.16 V$_{RHE}$ compared with that of the slow-cooling counterpart (0.09 V$_{RHE}$), indicating that the fast-cooling strategy is advantageous for not only the photocurrent but also the photovoltage and fill factor. As shown in Fig. 2d, both fast- and

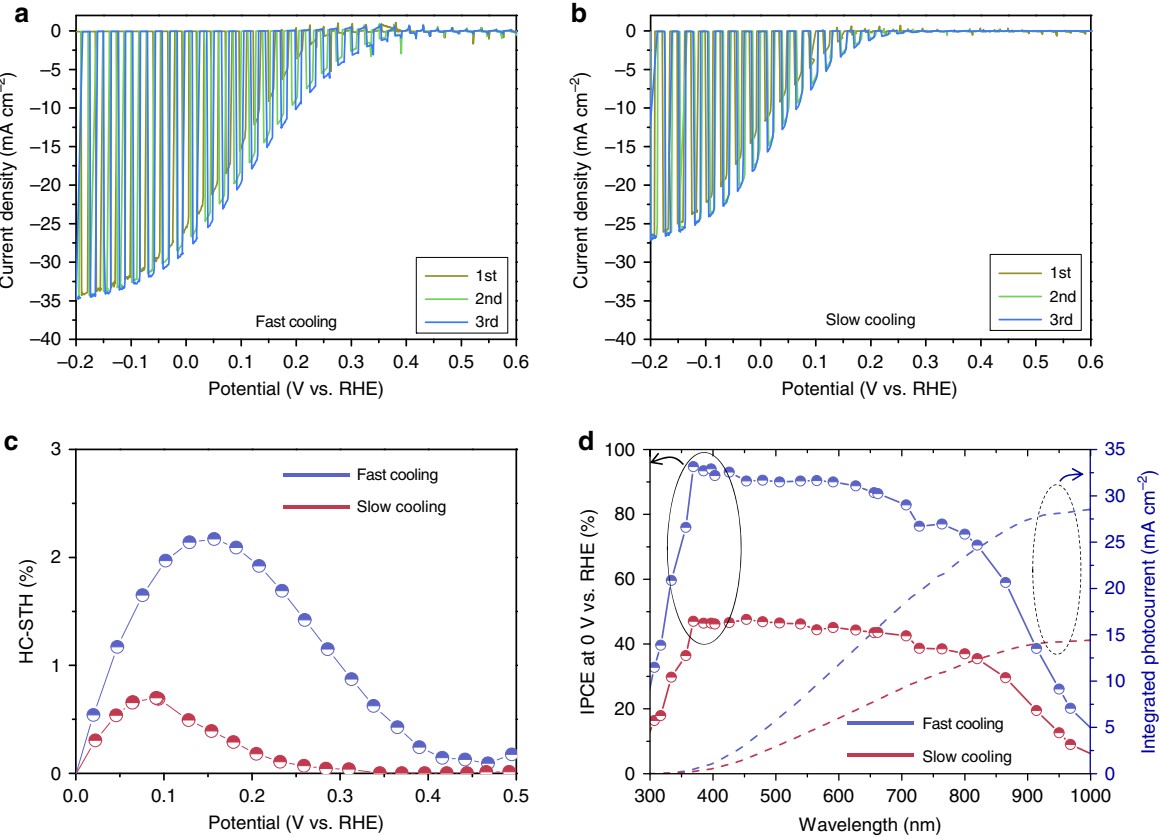

**Fig. 2 PEC performance of RuO$_x$/TiO$_2$/Sb$_2$Se$_3$/Au/FTO photocathodes in pH 1 H$_2$SO$_4$ electrolyte. a, b** *J–V* curves of **a** fast- and **b** slow-cooling samples under simulated 1 sun air mass 1.5 G chopped illumination at a scan speed of 5 mV s$^{-1}$ from positive to negative potential and **c** corresponding HC-STH efficiencies. **d** Wavelength-dependent IPCE and integrated photocurrent density of fast- and slow-cooling samples measured at 0 V$_{RHE}$. Source data used to generate this figure can be found in the Source Data file.

slow-cooling Sb$_2$Se$_3$ photocathodes were able to harvest photons over 1000 nm, but the IPEC was much higher for the fast-cooling sample over the entire wavelength. Because the optical properties (e.g. surface reflection of Sb$_2$Se$_3$/Au/FTO samples) for the fast- and slow-cooling samples were nearly identical (Supplementary Fig. 7), optical properties can be excluded from the potential origin of the difference noted in the performance. In addition, a similar performance was observed when a Pt co-catalyst was used (Pt/TiO$_2$/Sb$_2$Se$_3$/Au/FTO, Supplementary Fig. 8): thus we can conclude that the performance difference arose not from the co-catalyst but presumably from the TiO$_2$/Sb$_2$Se$_3$ junction.

In order to understand the performance difference observed for the fast- and slow-cooling samples, we performed Kelvin probe force microscopy (KPFM) analyses, which allowed us to investigate the topography and surface potential distribution. Figure 3a, c and b, d show the surface topography and surface potential of TiO$_2$/ Sb$_2$Se$_3$ devices, respectively. The fast-cooling TiO$_2$/Sb$_2$Se$_3$ revealed relatively uniform distribution of surface topography and potential, as shown in Fig. 3a, b. On the other hand, some noticeable high-potential regions, as represented by the bright yellow colour in Fig. 3d, were observed for the slow-cooling TiO$_2$/Sb$_2$Se$_3$ sample. Topography and potential line profiling for the fast-cooling sample suggested that the potential resembles the topography of TiO$_2$/ Sb$_2$Se$_3$ (Fig. 3e), indicating a lower surface potential at the grain boundaries of Sb$_2$Se$_3$[35]. In the fast-cooling sample, the separation of the photo-generated electrons and holes efficiently occurred along the vertically aligned [Sb$_4$Se$_6$]$_n$ ribbons, as shown in Fig. 3g, owing to the p–n junction and relatively small lateral potential difference. In contrast, in the slow-cooling sample, the surface potential increased significantly with a rapid drop in the topography (Fig. 3f),

indicating direct contact between the n-type TiO$_2$ layer and substrate due to pin-holes. It might be worth to note that the height recorded by AFM and KPFM ranged from a few nm to hundreds nm and even sometimes μm scale[36,37]. In such a case, the photo-excited electrons can be extracted laterally to the ribbons and they can recombine with the holes at the back contact as shown in Fig. 3h due to the large electric field across the p–n junction. It is widely known that direct contact between the top and bottom contact (i.e., in this case, the TiO$_2$ and Au layers) can cause significant degradation of the performance, even in the case of the chemical composition and optoelectronic properties have negligible differences. For example, Luo et al. reported the effect of a thin blocking layer to prevent shunt pathways, thereby enabling much higher performance in Cu$_2$O nanowire photocathodes without any noticeable differences in chemical composition and morphology[38]. The KPFM results clearly demonstrated the importance of pin-hole-free compact thin films in preventing the recombination and the performance degradation mechanism in the presence of pin-holes, which had not yet been experimentally demonstrated.

**Sb$_2$Se$_3$ photocathodes in unassisted water splitting.** Despite the high photocurrent density of the fast-cooling CSS-Sb$_2$Se$_3$ photo-cathodes, there remain some issues prior to achieving unassisted water splitting by fabricating a tandem device in conjunction with the photoanode. After the optimisation of the electrolyte and the onset potential (Supplementary Figs. 9–11), we decided to use a Pt co-catalyst for the PEC tandem device. With the Pt co-catalyst, we inserted a CdS layer between TiO$_2$ and the fast-cooling Sb$_2$Se$_3$ to further enhance the onset potential, as shown in Fig. 4a[27]. The

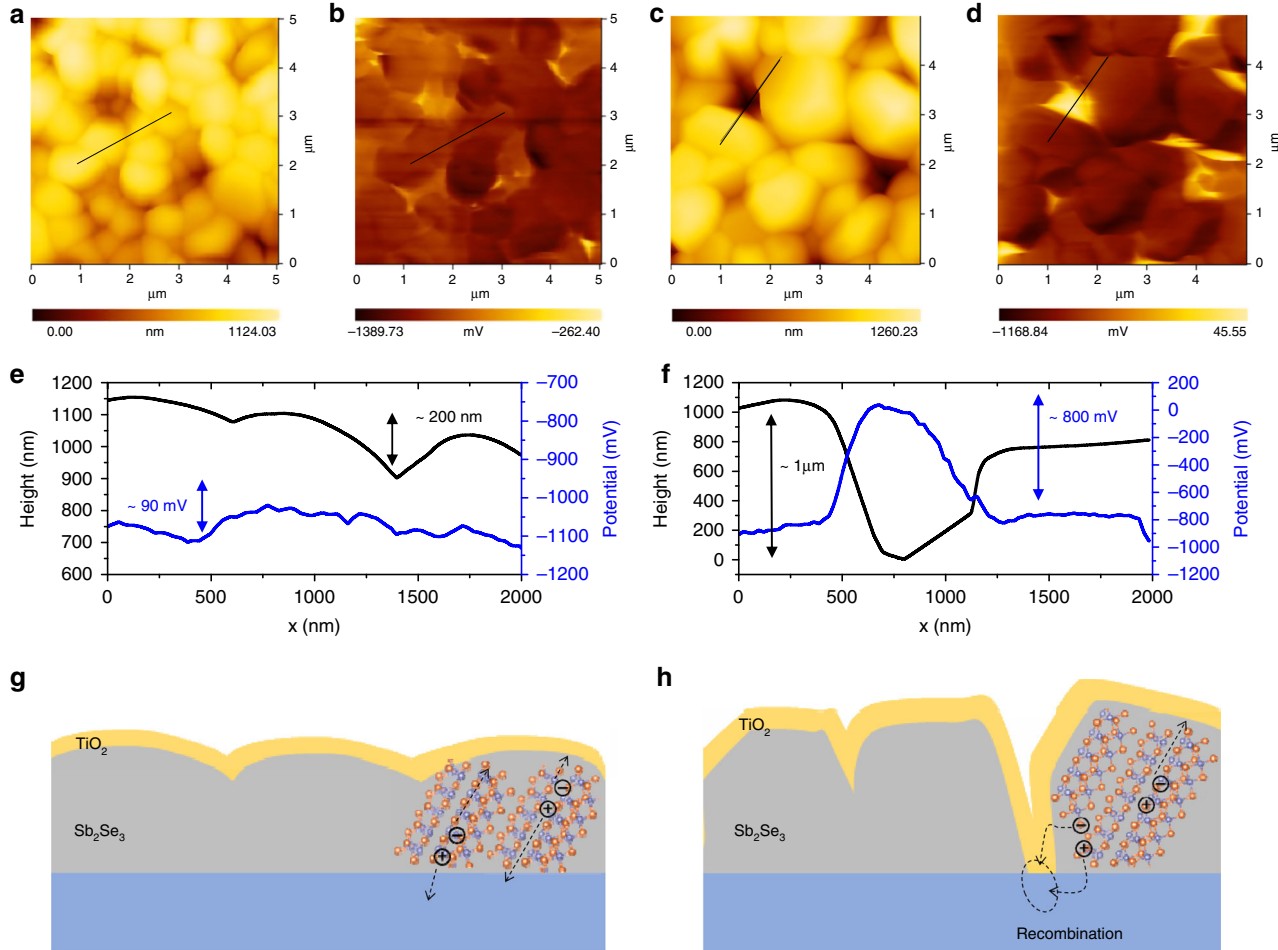

**Fig. 3 Topography and surface potential of TiO₂/Sb₂Se₃ according to KPFM analysis. a**, **c** Topography of **a** fast- and **c** slow-cooling samples. **b**, **d** Surface potential of **b** fast- and **d** slow-cooling samples. **e**, **f** Topography and potential line profile (obtained from scanning the black lines in Fig. 3a–d) for **e** fast- and **f** slow-cooling samples. **g**, **h** Schematics showing possible charge separation and recombination in **g** fast- and **h** slow-cooling samples. Source data used to generate this figure can be found in the Source Data file.

thickness of the CdS and TiO₂ layers were approximately 50 and 30 nm, respectively (Supplementary Fig. 12). Upon deposition of the CdS layer, the onset potential of the Sb₂Se₃ photocathode shifted up to 0.5 $V_{RHE}$ and the fill factor was also significantly enhanced (Fig. 4b). As a result, the maximum HC-STH increased from 2.33% to 3.4%, and the potential at which the maximum HC-STH was observed also shifted from 0.18 to 0.26 $V_{RHE}$, as shown in Fig. 4c. Therefore, despite the slightly decreased photocurrent density at 0 $V_{RHE}$, Pt/TiO₂/CdS/Sb₂Se₃ configuration was more suitable for the PEC tandem device. The enhanced onset potential and fill factor could be attributed to the reduced band mismatch between TiO₂ and Sb₂Se₃[27]. Figure 4d shows the IPCE spectra for Sb₂Se₃ photocathodes with and without CdS layers, suggesting that the IPEC of 300–500 nm decreased with the thickness of the CdS layer. The reduced IPCE values imply that the photons absorbed by the CdS ($E_g$ ~ 2.4 eV, ~516.6 nm) were unable to contribute to the photocurrent density. Therefore, when the thickness of the CdS layer increased, the photocurrent density decreased, whereas the onset potential remained the same (i.e. ~0.5 $V_{RHE}$), indicating the necessity of careful optimisation of the CdS layer thickness (Supplementary Fig. 13).

**D4 tandem cell for unbiased solar overall water splitting.** Since the band gap of Sb₂Se₃ is 1.2 eV, which is suitable for realising a D4 tandem cell as a bottom electrode with a different top light

absorber[39], we fabricated tandem cells coupled with BiVO₄-based photoanodes ($E_g$ ~ 2.4 eV), which have been widely used for high-performance photocathode–photoanode tandem cells[10] (Fig. 5a). The H,Mo:BiVO₄ photoanode used here revealed good transparency (Supplementary Fig. 14) and performance (~4.7 mA cm⁻² at 1.23 $V_{RHE}$ and onset potential of 0.25–0.3 $V_{RHE}$) as a top light absorber. To achieve high efficiency, further optimisation of the electrolyte was conducted for the tandem cell comprising two photoelectrodes involving opposite reactions (Supplementary Figs. 15–18).

As shown in Fig. 5b, the operating point of the two photoelectrodes, as estimated by the intersection of two J–V curves, was approximately 1.2 mA cm⁻² at 0.4 $V_{RHE}$, which corresponded to a STH efficiency of 1.5%. It should be noted that photocurrent density of BiVO₄ at 0.4 $V_{RHE}$ varies from 1.2 to 0.8 mA cm⁻², thus the overall STH efficiencies of the tandem cell range from 1.48% to 0.98%. In addition, the polarisation curves of BiVO₄–Pt (as the counter electrode) and BiVO₄–Sb₂Se₃ (as the counter electrode) demonstrated a ~0.5 V anodic shift induced by the Sb₂Se₃ photocathode, indicating that the Sb₂Se₃ photocathode provided a photovoltage of 0.5 V (Fig. 5c). The BiVO₄–Sb₂Se₃ tandem cell exhibited 1.2 mA cm⁻² at 0 V against the counter electrode (i.e. unbiased conditions), which was in agreement with the estimated value according to the overlapped J–V curves shown in Fig. 5b. This observation verified that the total photovoltage of 1.45 V (0.52 V from the Sb₂Se₃ photocathode

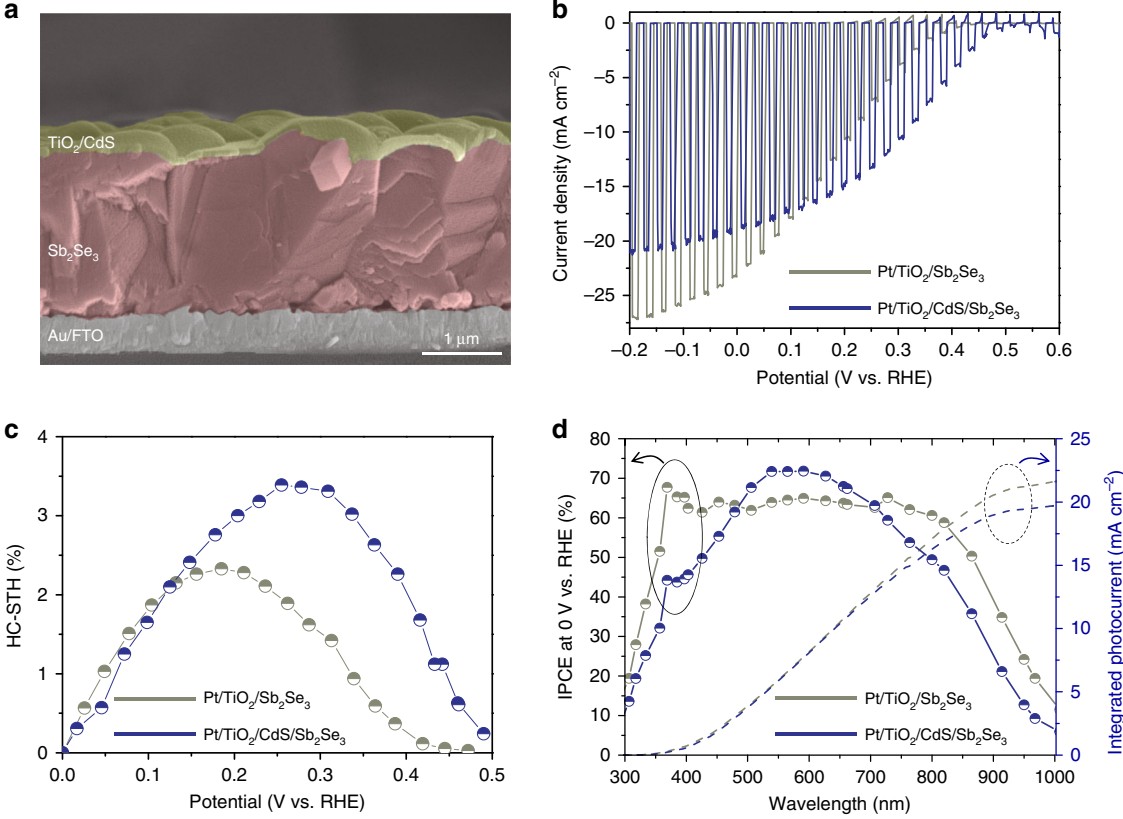

**Fig. 4 Microstructure and PEC performance of Pt/TiO$_2$/CdS/Sb$_2$Se$_3$/Au/FTO photocathodes. a** Cross-sectional SEM image of TiO$_2$/CdS/Sb$_2$Se$_3$ on Au/FTO substrate. **b** J–V curves for Sb$_2$Se$_3$ photocathodes with/without a CdS layer in pH 1 H$_2$SO$_4$ under simulated 1 sun air mass 1.5 G chopped illumination at a scan speed of 5 mV s$^{-1}$ from positive to negative potential and **c** corresponding HC-STH efficiencies. **d** Wavelength-dependent IPCE at 0 V$_{RHE}$ of Sb$_2$Se$_3$ photocathodes with/without CdS layer. Source data used to generate this figure can be found in the Source Data file.

and 0.93 V from the BiVO$_4$ photoanode) was sufficient to implement unbiased overall water splitting, resulting in evolution of both H$_2$ and O$_2$, by overcoming the thermodynamic potential requirement (1.23 V). In comparison to previously reported photoelectrodes, the total photovoltage of 1.45 V was a moderate value considering the band gap energies[40]. Thus it is noteworthy that a photovoltage increment of ~0.1 V and the enhanced fill factor obtained using CdS played a critical role in driving the overall water splitting.

Photon utilisation capability of the BiVO$_4$–Sb$_2$Se$_3$ tandem cell was confirmed by IPCE measurements. For BiVO$_4$, photons up to ~510 nm ($E_g$ of 2.45 eV) could be converted and the absorption current density, which represents the maximum photocurrent density under the absorbance of BiVO$_4$, was calculated to be 5.5 mA cm$^{-2}$ (Supplementary Fig. 14b). Sum of the photon flux before and after passing through BiVO$_4$ up to the wavelength of 1033 nm (~1.2 eV, $E_g$ of Sb$_2$Se$_3$) was calculated to be 39.9 and 24.4 mA cm$^{-2}$, respectively, which are much higher than the photocurrent of BiVO$_4$ at the top of the tandem cell configuration.

We compared BiVO$_4$ and Sb$_2$Se$_3$ in two different bias conditions: high bias near short circuit of the individual photoelectrode (1.0 V$_{RHE}$ for photoanode and 0.1 V$_{RHE}$ for photocathode; Supplementary Fig. 19) and operating bias at 'actual' operating point (0.4 V$_{RHE}$ for both the photocathode and photoanode; Fig. 5d). In both cases, a broad range of photons could be effectively utilised by the synergetic effect of BiVO$_4$ (to 510 nm) and Sb$_2$Se$_3$ (from 450 to 1000 nm). In case of the high bias condition, both the photocathode and photoanode showed a quite high IPCE (~70% at 450 nm for BiVO$_4$ and ~60%

for Sb$_2$Se$_3$). The photocurrent calculated from the IPCE was 3.2 mA cm$^{-2}$ for the BiVO$_4$ photoanode. In addition, the photocurrent of the Sb$_2$Se$_3$ photocathodes was 18.5 and 10.3 mA cm$^{-2}$ with or without BiVO$_4$, respectively. These values are in a good agreement with photocurrents shown in the J–V curves (Supplementary Fig. 18). On the other hand, as expected, a lower IPCE was observed under the operating bias (0.4 V$_{RHE}$): ~25% at 450 nm for BiVO$_4$ and ~15% for Sb$_2$Se$_3$, as shown in Fig. 5d. The photocurrent density calculated using the IPCE spectra of BiVO$_4$ at 0.4 V$_{RHE}$ was about 1.1 mA cm$^{-2}$, which is similar to the values shown by the J–V curve (Supplementary Fig. 18). However, the photocurrent density of Sb$_2$Se$_3$ photocathodes at 0.4 V$_{RHE}$ with and without BiVO$_4$ were 5.0 and 2.2 mA cm$^{-2}$, respectively, which are higher than the photocurrents shown by the J–V curve. This discrepancy between the photocurrent density at 0.4 V$_{RHE}$ demonstrated by the IPCE and J–V curve can be attributed to the light-intensity-dependent carrier mobility of Sb$_2$Se$_3$. As the carrier mobility of Sb$_2$Se$_3$ is much higher under low-intensity light[32], Sb$_2$Se$_3$ photocathodes showed a better fill factor in IPCE measurement conditions (~100 μW cm$^{-2}$) compared with the 1 sun conditions (100 mW cm$^{-2}$), thereby resulting in a higher photocurrent according to the IPCE spectra at 0.4 V$_{RHE}$. The D4 tandem cell consisting of Sb$_2$Se$_3$ and BiVO$_4$ operated stably for over 10 h without any noticeable degradation in the photocurrent density (Fig. 5e). The efficiency and stability of our Sb$_2$Se$_3$-based tandem cells are comparable to those of the best-performing tandem cells containing other types of photocathode materials (vide infra). Finally, a larger tandem cell was constructed using a BiVO$_4$ photoanode (0.81 cm$^{-2}$) and a Sb$_2$Se$_3$ photocathode (0.35 cm$^{-2}$)

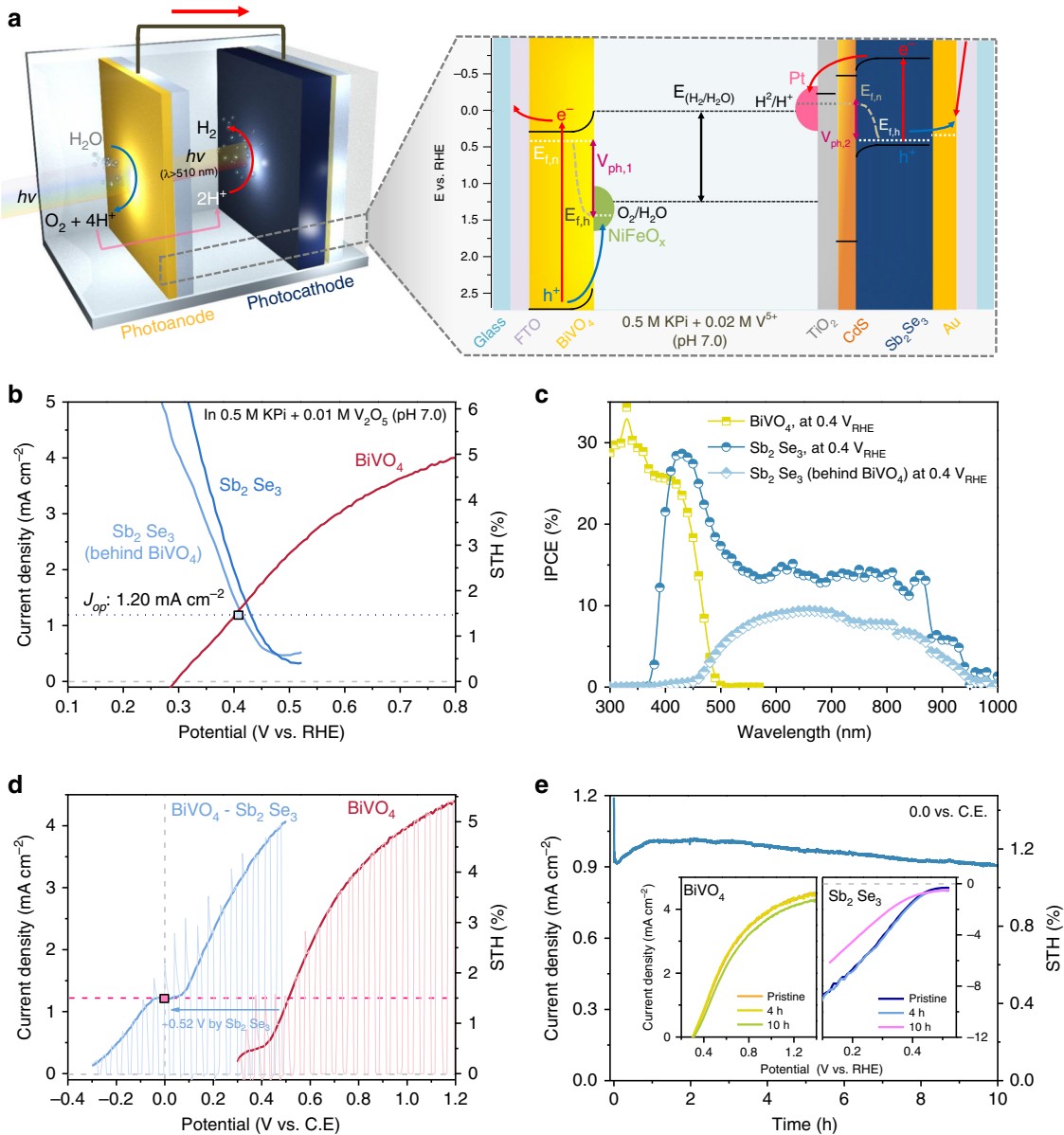

**Fig. 5 Sb$_3$Se$_3$ photocathode–BiVO$_4$ photoanode for solar overall water splitting PEC cell. a** Scheme of the NiFeO$_x$/H,Mo:BiVO$_4$/FTO-Pt/TiO$_2$/CdS/ Sb$_2$Se$_3$/Au/FTO tandem cell operating in pH 7.0 phosphate buffer. **b** J–V curve for the photocathode and photoanode; the operating point is marked for the tandem cell (photoelectrode active area: 0.32 cm$^2$). **c** Two-electrode measurements for photoanode–counter electrode (C. E.) and photoanode–photocathode tandem cells (active area: 0.28 cm$^2$). **d** IPCE at 0.4 V$_{RHE}$ measured for the photoanode and photocathodes. **e** Photocurrent generation under short circuit conditions (0 against counter electrode) of the photoanode–photocathode tandem cell. All analyses were conducted in 0.5 M phosphate buffer + 0.01 M V$_2$O$_5$ (pH 7.0) at a scan rate of 20 mV s$^{-1}$. Distance between the photoanode and photocathode was ~0.5 cm. Source data used to generate this figure can be found in the Source Data file.

under simulated ~3 sun, which clearly demonstrated a constant generation of O$_2$ and H$_2$ gas evolution (Supplementary video 1, Supplementary Fig. 20).

## Discussion

Even though we report herein the first demonstration of unbiased water splitting using a Sb$_2$Se$_3$ photocathode in a single reactor, better tandem cell systems are likely achievable by using two different electrolytes that are separately optimised for each photoelectrode (e.g. pH 1 for Sb$_2$Se$_3$ and pH 9 borate for BiVO$_4$) along with a bipolar membrane[41] or stabilising both the photocathode and photoanode using additives (e.g. chelating agent to deter poisoning) in the electrolyte[42]. Bearing in mind its short history

and potential for further enhancement, the performance of the Sb$_2$Se$_3$-based tandem cell (STH efficiency of 1.5% and stability of over 10 h) is quite remarkable particularly in comparison with previously reported results for photocathode–photoanode tandem cells[4] (Fig. 6a, Table 1 in Supplementary Information). We categorised the tandem cells based on the bottom light absorbers (silicone based, III–V, Cu-chalcogenides, metal oxides, and perovskites). Most of them used BiVO$_4$, which is currently the most suitable photoanode material and ensures good transparency (~80% beyond photon utilisation threshold (~520 nm))[10]. It was found that the efficiency of the BiVO$_4$–Sb$_2$Se$_3$ tandem cell (1.5% STH) was not the highest when compared to that of BiVO$_4$–CIGS (3.7%)[43], BiVO$_4$–Cu$_2$O (3.0%)[44], and BiVO$_4$–c-Si (2.1%)[41]. Considering the cost of materials (much lower than that of CIGS),

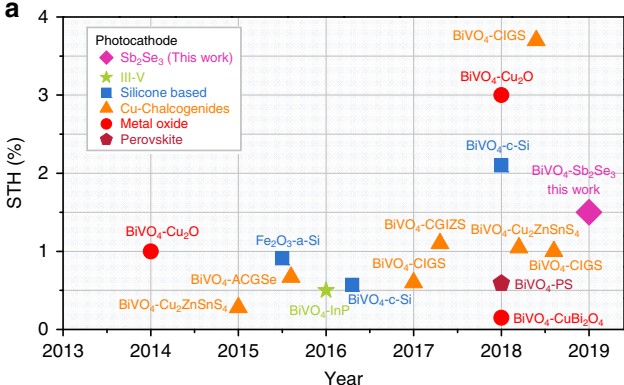

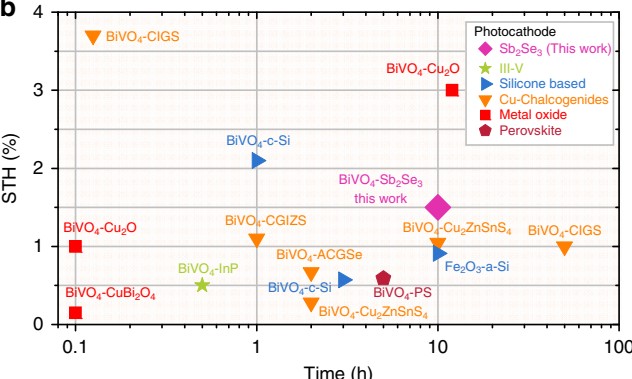

**Fig. 6 Efficiency and stability benchmarks for photoanode–photocathode D4 tandem cell. a** Solar-to-hydrogen efficiencies reported in recent years and **b** their operation duration. All reports are summarised in Supplementary Table 1.

small band gap (compared to that of $Cu_2O$, $E_g \sim 2.0$ eV), simple preparation, and low material usage (compared to that of Czochralski-grown Si wafer and complicated Si microwires), the >1% STH can be considered a significant milestone for PEC tandem cells. Moreover, 10 h of stability is also the best result reported for photoanode–photocathode tandem cells except for the 20 h stability achieved by the expensive CIGS-based photocathodes[45] (Fig. 6b). The exceptional performance and stability of the $Sb_2Se_3$ photocathode presented here, as well as several desirable material characteristics of $Sb_2Se_3$ such as its high $\alpha$, optimal band gap of ~1.2 eV, and comparable photovoltage to that of c-Si (~0.5 V), clearly present a promising pathway towards realising a competitive overall water splitting PEC cell system, which could potentially result in an STH of approximately 10% using $BiVO_4$ photoanodes whose $E_g$ is 2.4 eV (Supplementary Fig. 21a). In order to demonstrate a higher STH efficiency, the areas of focus should include enhancing both the photovoltage and fill factor through interface engineering, such as the recently reported dipole engineering at the buried junction[46]. In addition, the development of an ideal photoanode material with an $E_g$ close to 1.8 eV could potentially enable an STH efficiency of >20% on using the $Sb_2Se_3$ photocathode (Supplementary Fig. 21b). Although the value of STH 1.5% is not superior in comparison to that of other systems, such as photovoltaic–electrolytic systems (average 10–20%), buried-junction photoelectrodes (10–15%), and photovoltaic–PEC systems (3–8%), because of the simplicity of the photoanode–photocathode system, the price of hydrogen produced from such a system will be lower. Therefore, not only the STH efficiency but also the cost should be considered when comparing such systems. The reported STH efficiencies of the photoanode–photocathode

tandem system are increasing very rapidly (Fig. 6), which suggests increasing research interest in reducing the complexity of overall water splitting devices[4].

In summary, we have demonstrated a high-performance photocathode using a low-cost $Sb_2Se_3$ semiconductor, which has a small $E_g$ (~1.2 eV), good optoelectronic properties, and no secondary phase. By suppressing the growth kinetics during the CSS deposition, a compact and pinhole-free $Sb_2Se_3$ thin film was obtained, which enabled a highest photocurrent density of up to 30 mA cm$^{-2}$ by avoiding recombination and accelerating charge separation through well-aligned $(Sb_4Se_6)_n$ ribbons. Inserting a CdS layer between $Sb_2Se_3$ and $TiO_2$ increased the onset potential and consequently improved the HC-STH efficiency up to 3.4%. Finally, by combining the material with a $BiVO_4$ photoanode, unbiased overall water splitting was achieved with impressive efficiency (~1.5%) and high stability of 10 h. These performances and stability significantly surpass those of previously reported $Sb_2Se_3$ photocathodes and are comparable to those of other expensive thin film photocathodes. Given the relatively short history of photoelectric materials, it is expected that rapid progress of $Sb_2Se_3$ photocathodes will lead to better efficiency in the near future and our findings represent an important demonstration of a photocathode–photoanode-based PEC device. We believe that the emerging $Sb_2Se_3$ substrate can be an attractive breakthrough material for practical solar fuel production.

## Methods

**Preparation of $Sb_2Se_3$ thin films.** $Sb_2Se_3$ films were deposited via a two-step CSS process[31] using a custom-made CSS system. First, a compact layer was grown from stoichiometric $Sb_2Se_3$ (Alfa Aesar, 99.999% metals basis) using a source temperature of 340 °C and a substrate temperature of 390 °C for 2 min at a pressure of 0.05 mbar. Subsequently, deposition was completed using a source temperature of 460 °C for 15 min and a pressure of 13 mbar. These films were then cooled either slowly or fast, by turning the heater off and flowing $N_2$ over the sample at a rate of either 0 or 5 L min$^{-1}$, respectively.

**Deposition of overlayers and catalysts for photocathodes.** CdS layers were deposited by the chemical bath deposition (CBD) method. Prior to CBD, the sample was pre-treated in a bath containing a solution of $CdSO_4$ (Sigma Aldrich, 99.99%) and $NH_4OH$ (Duksan, 28 wt%) at 60 °C for 10 min. CBD of CdS was performed by immersing the pre-treated sample in a solution containing $CdSO_4$, thiourea (99%, Sigma Aldrich), deionised (DE) water, and $NH_4OH$ for 5 min at 60 °C. $TiO_2$ layers were deposited using an ALD system (Lucida D100, NCD Inc.). The ALD process was performed at 150 °C, with tetrakis(dimethylamido)titanium (TDMAT) and $H_2O$ as the Ti and O sources, respectively. A total of 600 ALD cycles were carried out, each of which comprised a TDMAT pulse of 0.3 s followed by 15 s of $N_2$ purging and a $H_2O$ pulse of 0.2 s followed by 15 s of $N_2$ purging. The approximate growth rate of $TiO_2$ was 0.55 Å per cycle, as estimated using an ellipsometer. The Pt catalyst was sputtered onto the $TiO_2$-coated $Sb_2Se_3$ electrode using an Auto Sputter Coater (Ted Pella, Redding, CA, USA) under an applied current of 10 mA for 120 s. A galvanostatic photo-deposition technique was used for $RuO_x$ deposition. The prepared photoelectrodes were immersed in a 1.3 mM solution of $KRuO_4$ at a current density of $-28.3$ μA cm$^{-2}$ for 15 min under simulated 1 sun illumination.

**Preparation of $BiVO_4$ films.** All chemicals used in this study were of analytical grade and used without further purification. $BiVO_4$ film was prepared by a modified metal–organic decomposition method according to our previously reported procedure[47] with slight modifications. In brief, 291 mg of $Bi(NO_3)_3\cdot5H_2O$ (99.8%; Kanto Chemicals) and 163 mg of $VO(acac)_2$ (98.0%; Sigma Aldrich) were dissolved in 15 mL of acetyl acetone (>99.0%; Kanto Chemicals). As the dopant solution, 0.03 M $MoO_2(acac)_2$ (98.0%; Sigma Aldrich) in acetyl acetone (>99.0%; Kanto Chemicals) was prepared. A 0.2 M $Bi(NO_3)_3\cdot5H_2O$ (99.8%; Kanto Chemicals) solution dissolved in acetic acid (>99.0%; Kanto Chemicals) was used to compensate the Mo precursor-added solution. For Mo doping, a Bi:(V+Mo) = 1:1 atomic ratio was used for preparing the 1% Mo:$BiVO_4$ films. To fabricate the $BiVO_4$ film, 35 μL of the solution was dropped on a FTO glass sample (2 cm × 2.5 cm) and dried for 10 min in Ar atmosphere. The FTO glass (TEC 8; Pilkington) was cleaned using ethanol +35 wt% $H_2O_2$ (Samchun Chemical) used in a ratio of 5:1 under sonication (~20 min) and washed with copious amounts of ethanol and finally stored in 2-propanol before use. The greenish transparent precursor film was calcined at 550 °C for 30 min to form a yellow $BiVO_4$ film. After the annealing process, the 2 cm × 2.5 cm $BiVO_4$/FTO was split to obtain photoanodes with a net

irradiation area of $0.36 \text{ cm}^2$ connected by silver paste and copper wire and sealed with epoxy resin.

**Hydrogen treatment of metal oxide films.** Hydrogen treatment was conducted using the borohydride decomposition method reported by Hao et al.[48]. First, 16 mmol of $NaBH_4$ (>98%; Sigma Aldrich) was introduced in a 200-mL alumina crucible and another smaller alumina bottle (15 mL) was introduced to the $NaBH_4$ powder. In this smaller bottle, an as-prepared metal oxide film (2 cm × 2.5 cm) was placed and finally the 200-mL alumina crucible was covered with an alumina cover. This reactor was placed in an already heated furnace at 500 °C for 30 min. Then the crucible was taken out from the furnace and cooled naturally.

**NiFeO$_x$ co-catalyst deposition on BiVO$_4$ film.** The $NiFeO_x$ co-catalysts were deposited by photo-assisted electrodeposition under air mass (AM) 1.5 G illumination according to a reported procedure[49]. In all, 60 mg of $Fe(SO_4)_2 \cdot 7 H_2O$ (≥99%; Sigma Aldrich) and 20 mg of $Ni(SO_4)_2 \cdot 6H_2O$ (99%; Sigma Aldrich) were put in a glass bottle and 200 mL of 0.5 M $KHCO_3$ (pH 8.3, purged with Ar gas for 30 min before use) was introduced, resulting in a transparent and yellow solution. For deposition, under illumination (AM 1.5 G, $100 \text{ mW cm}^{-2}$), linear sweep voltammetry was conducted with a bias of −0.3 to 0.5 V against the reference electrode (Ag/AgCl) for 12–15 times with pre-treatment of −0.3 V for 5 s. Sequential linear sweep voltammetry resulted in reduced current density and over-deposition of $NiFeO_x$. After deposition, the photoelectrode was taken out and washed with copious amounts of DI water. Right after being taken out, the photoelectrode showed a slightly darkened colour ($Ni(OH)_2$ species) but gradually changes to a colour identical to that of the photoelectrode before $NiFeO_x$ deposition. The photoelectrodes were stored in an Ar gas-filled bottle before use.

**Characterisation.** The surface morphology of $Sb_2Se_3$ thin films were analysed via field-emission scanning electron microscope (JSM-7001F, JEOL Ltd, Tokyo, Japan). The crystallinity of the samples was investigated using XRD (MiniFlex 600, Rigaku, Tokyo, Japan) with Cu Kα radiation ($\lambda = 1.54178$ Å). The surface reflectance of the $Sb_2Se_3$ thin films was measured using an ultraviolet–visible (UV-vis) spectrophotometer (V-670, JASCO, Easton, MD, USA). UV-Vis absorbance of $BiVO_4$ was recorded with a UV/Vis spectrometer (UV-2401PC, Shimadzu). As a reference, $BaSO_4$ powder was used. In addition, KPFM (SPA 400, Seiko Instrument Inc., Chiba, Japan) measurements were performed using a gold-coated cantilever (SI-DF3-A).

**Determination of PEC performance.** PEC measurements for $Sb_2Se_3$ photocathodes were performed in a typical three-electrode system with a Ag/AgCl/KCl (4 M) reference electrode and a Pt wire counter electrode. The $Sb_2Se_3$ photocathodes were submerged in an acidic ($H_2SO_4$, pH ~ 1) or a neutral (phosphate buffer, pH ~ 6.25) electrolyte and simulated solar light illumination. Calibration of the 1-sun level was performed using a standard Si reference cell certified by the Newport Corporation, consisting of a readout device and a 2 × 2 cm$^2$ calibrated solar cell made of monocrystalline silicon. During calibration, the Si reference cell was located at the same position of the sample for PEC measurement. The scan rate for the J–V curves was $5 \text{ mV s}^{-1}$. For $Sb_2Se_3$ photocathodes, the applied potentials were recorded against the RHE to allow comparison with previously reported results, employing the relationship $E_{RHE} = E_{Ag/AgCl} + 0.059 \text{ pH} + 0.197$. IPCE for $Sb_2Se_3$ photocathodes was measured with an electrochemical workstation (Zennium, Zahner, Germany) combined with a potentiostat (PP211, Zahner, Germany) under monochromatic light.

PEC performance of the $BiVO_4$ photoanode was measured with a photoanode as the working electrode, Pt mesh as the counter electrode, and Ag/AgCl (3 M NaCl) as the reference electrode. The scan rate for the J–V curves was $20 \text{ mV s}^{-1}$. For electrolyte, 0.5 M potassium phosphate ($K_2HPO_4$ or KPi) buffer + 0.01 M $V_2O_5$ (pH 7.0) was used as a standard electrolyte as reported previously for the borate buffer (pH 9.5). To accelerate dissolution of $V_2O_5$, it was placed in oven at 80 °C for 3 h. The pH of the KPi buffer barely changed (<0.01) while the pH of 8 or 9 showed a noticeable change that could be compensated by adding a small amount of 0.5 M KOH. Potentials were recorded with correction according to the Nernst relationship $E_{RHE} = E_{Ag/AgCl} + 0.0591 \text{ pH} + 0.209$, in which $E_{Ag/AgCl}$ is the applied bias potential and 0.209 is a conversion factor from the Ag/AgCl electrode to the RHE scale. The electrochemical data for $BiVO_4$ photoanodes were recorded by using a potentiostat (IviumStat, Ivium Technologies). A 300 W Xenon lamp was used to produce simulated 1 sun light irradiation conditions (AM 1.5 G, 100 mW $cm^{-2}$) employing a solar simulator (Oriel 91160) with an AM 1.5 G filter calibrated with a reference cell certified by the National Renewable Energy Laboratories, USA.

For the $BiVO_4$–$Sb_2Se_3$ tandem cell, black masked (0.32–0.36 cm$^2$) photoelectrodes were aligned with a spacing of ~0.5 cm in a single electrolyte bed reactor. Photocurrent of $Sb_2Se_3$ was recorded with or without the $BiVO_4$ photoelectrode (fully modified, $NiFeO_x$/H,Mo:$BiVO_4$) under backward scan (~0.5–0 $V_{RHE}$). The $BiVO_4$ photoelectrode was scanned in the forward scan (0.2–1.4 $V_{RHE}$). For the $BiVO_4$–$Sb_2Se_3$ two-electrode full cell, $BiVO_4$ was set as the working electrode and $Sb_2Se_3$ was set as the counter electrode. Bias was applied towards the counter electrode and thus 0 V against the counter electrode indicated a 0 V potential applied to the full cell. For linear sweep, forward scan was applied for

the $BiVO_4$–Pt rod (0.3–1.3 V) or $BiVO_4$–$Sb_2Se_3$ (−0.3 to 0.5 V). All scans were performed with 10 s of pre-treatment at the initial potential at a speed of $20 \text{ mV s}^{-1}$.

IPCE measurements for $BiVO_4$ and $Sb_2Se_3$ behind $BiVO_4$ were conducted using a 300 W Xe lamp as the light source with a liquid infrared filter and a monochromator (Oriel Cornerstone 130 1/8 m monochromator) with a bandwidth limit of 5 nm. The intensity of light was measured before IPCE measurements by a photodiode detector (Oriel 70260). Calculation of IPCE was carried out according to the equation:

$$IPCE(\%) = \frac{1240 \times J}{\lambda \times P} \times 100 \tag{1}$$

where $J$ = photocurrent density (mA cm$^{-2}$), $P$ = light power density (mW cm$^{-2}$) at $\lambda$, and $\lambda$ = wavelength of incident light (nm). $J$ was gathered under the condition that the PEC cell individually installed and the constant potential applied for a photoelectrode and wavelength of illuminated beam (1 cm × 1 cm) was periodically changed. The active area of the characterised photoelectrode was preferably smaller than this beam source (0.32–0.36 cm$^2$). The scanning wavelength range was 300–1100 nm with an interval of 10 nm/7 s.

## Data availability
The data that support the plots within this paper and other findings of this study are available from the corresponding author upon reasonable request. The source data underlying Figs. 2, 3e, f, 4b–d, 5b–e, and 6 and Supplementary Figs. 2, 3, 4, 5c, 7, 8, 9, 10, 11a, 13, 14, 15, 16, 17, 18b–d, 19, and 21a are provided as a Source Data file.

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

## Acknowledgements

This work was supported by a National Research Foundation (NRF) of Korea grant (No. 2012R1A3A2026417); a Creative Materials Discovery Program (NRF-2018M3D1A1058793) funded by the Ministry of Science and ICT; the Korea Center for Artificial Photosynthesis (KCAP, No. 2009–0093880), funded by the MSIT; Project No. 10050509 funded by the MOTIE of Republic of Korea; and the UK EPSRC project number EP/N014057/1.

## Author contributions

W.Y. conceived the idea, organised the collaboration, conducted experiments, analysed the data, and wrote the manuscript. J.H.K. prepared the BiVO$_4$ photoanodes, tested the tandem device, and co-wrote the manuscript. O.S.H. and L.J.P. prepared and optimised the Sb$_2$Se$_3$ thin films via the CSS method. J.T., J.P. and H.L. assisted in device optimisation and data analysis for the Sb$_2$Se$_3$ photocathodes. J.D.M. helped to the idea and manuscript preparation. J.S.L. directed the research and contributed to the writing of the manuscript. J.M. designed and supervised the project, directed the research, and contributed to the writing of the manuscript.

## Competing interests

The authors declare no competing interests.
