## [Peer Review File · Nature Communications]

Reviewers' comments:

Reviewer #1 (Remarks to the Author):

The authors reported a benchmark photocurrent 30 mA cm^{-2} at 0V RHE for the Sb_2Se_3 photocathode for the solar water splitting. The demonstrated 1.5% solar to hydrogen efficiency with stability over 10 h under 1sun also pave a way for the commercial solar fuel production. This work should be published in Nature Communications by addressing the following concerns.

1. The authors claim that improved PEC performance originates from the fast quench of the CSS process. However, the cooling rate comparison did not provide. Also, how fast the cooling rate? and how the cooling rate impacts the films chemical composition, grain size should be provided.
2. It is surprising that the IPCE improved from 45% to 95% in the short wavelength after fast cooling (Fig. 2d). Is there any optical transmittance for the substrate, Au coated FTO substrate. How the Au deposited on the FTO and how thick the Au layer? As mentioned by the authors, the optical properties for the fast and slow cooling samples are nearly identical. As the authors proposed, TiO_2 layer may contribute to the improved performance, however, the CSS can directly grow dense Sb_2Se_3 film (may cite Sol. RRL, 2: 1800128.2018 and Nano Energy, 49, 346, 2018). It is still unclear why the TiO_2 could boost the device performance significantly. In addition, in the KPFM images, the Fig.3c and 3f shown the surface profile. The KPFM tip may not record the 1 μm valley (till the FTO substrate?)
3. Again, the role of the Au layer on the FTO on the PEC performance is unclear, it should compare the sample with and without Au layer. Here, the Sb_2Se_3 grown on the FTO.
4. How the cells chemical composition post reliability test?

Reviewer #2 (Remarks to the Author):

The manuscript by Yang et al. describes a new synthesis method for Sb_2Se_3 photocathodes—a highly promising material for practical water splitting—and the development of a tandem cell with BiVO_4 . The performance of the Sb_2Se_3 is outstanding, the article is well written and appropriately cited, and the level of scientific discussion is high. I recommend to accept after addressing the following minor concerns:

The title should be more descriptive of the contents of the article.

Page 15: "The photocurrent density of $\text{Pt/TiO}_2/\text{Sb}_2\text{Se}_3$ decreased more rapidly (42% photocurrent after 5 h), probably due to the larger bubbles at Pt surfaces, as evidenced by severe fluctuations in the enlarged photocurrent curves (Supplementary Fig. 7b–c)." what do the authors mean, that the releasing of large bubbles cause the Pt to detach?

Page 18: "However, at a high concentration of 1.0 M, the stability of the photocathode degraded" Any thoughts why?

Page 20: "However, the photocurrent density of Sb_2Se_3 photocathodes at 0.4 VRHE with and without Sb_2Se_3 ..." with and without CdS ?

Fig 5 caption: "All analyses were conducted in 0.5 M phosphate buffer + 0.01 M V_2O_5 (pH 7.0)" – were faradaic efficiency measurements carried out to ensure that the 20 mM vanadium ions do not interfere with the hydrogen production? Since V^{5+} is much more easily reduced than proton.

Page 29: "and simulated solar light illumination (AM 1.5G, Newport Corporation) was used as the light

source." How was the intensity calibrated?

Reviewer #3 (Remarks to the Author):

The paper, "Benchmark performance of earth-abundant Sb₂Se₃ photocathodes for unassisted solar overall water splitting by Yang et al. presents a tandem Pt/TiO₂/CdS/Sb₂Se₃/Au/FTO photocathode and NiFeOx/BiVO₄/FTO photoanode that without a bias shows ~ 1 mA/cm² current density with ~ 1.1% STH using simulated AM1.5G 100 mW/cm² light.

There are no major concerns regarding the data or how the authors have conducted the experiments (although there are a couple of minor questions outlined below). The work is well done. However, there is little new science brought forth. Sb₂Se₃ modified with CdS and TiO₂ with at Pt catalyst has already been reported as a photocathode (see ACS Nano 2017, 11, 12753). The 1.1% sustained STH efficiency over 10 h is neither first nor best, and remains too low for practical water splitting. Admittedly the anode is the bigger problem, but it's not clear why this paper should be published in Nature Communications when many of the other references in Supplementary Table S1 are in a different tier of journals. This paper provides a few insights into materials engineering and device considerations, but even then, the ideas aren't new. Consequently, this paper is not recommended for publication in Nature Communications.

Minor questions about the data:

1. In the XRD shown in Figure 1, what would be the relative intensities of the (221) and (301) Bragg reflections in a truly randomly oriented sample? Is that what is shown in JCPDS 15-0861? The data for the fast and slow cooling look similar enough that I'm not convinced regarding the statement that a slight change in the XRD data implies that ribbons move during cooling.
2. In Figure 2a-b, the origin of the shift to more positive potentials after the 1st sweep is described as RuOx activation? Is the surface the same after this activation? What do the SEM images look like after HER?
3. Can the authors avoid mixing thermodynamic and kinetic language? Current is cathodic (or anodic), but there is no such thing as cathodic potential. So, rather than scanning in the "cathodic direction," the authors should state that J-V curves were scanned from open circuit to more negative potential.
4. Can the authors provide data for multiple films? It is not clear if the data are representative of typical films or the best-performing films.

Response Letter

Title: *“Benchmark performance of low-cost Sb₂Se₃ photocathodes obtained by the fast-cooling strategy during close space sublimation for unassisted solar overall water splitting”*

Referee #1

Comments to the Author

The authors reported a benchmark photocurrent 30 mAcm⁻² at 0V RHE for the Sb₂Se₃ photocathode for the solar water splitting. The demonstrated 1.5% solar to hydrogen efficiency with stability over 10 h under 1sun also pave a way for the commercial solar fuel production. This work should be published in Nature Communications by addressing the following concerns.

Author's response:

We thank the reviewer for evaluating our work as a good and outstanding work. All of the comments made by the reviewer are helpful for improving the overall quality of our work. Our detailed, point-by-point responses to the reviewer's comments can be found below.

Referee #1's Comment 1:

The authors claim that improved PEC performance originates from the fast quench of the CSS process. However, the cooling rate comparison did not provide. Also, how fast the cooling rate? and how the cooling rate impacts the films chemical composition, grain size should be provided.

Author's response:

We appreciate the reviewer for the helpful comment. We have measured the temperature of the close-space sublimation kit during cooling. While it fits nicely to a standard cooling curve, we feel it's clearer to approximate the initial temperature drop to a linear trend. As shown in Figure R1, the cooling rates for the fast cooling and the slow cooling until it has reached 200 °C are approximately 15.7 °C/min and 11.3 °C/min, respectively. Additionally, there was no noticeable difference in chemical composition of Sb₂Se₃ regardless of the cooling rate. The energy-dispersive X-ray spectroscopy (EDX) analysis showed that both fast and slow

cooling Sb_2Se_3 films are slightly selenium poor ($\text{Se}/\text{Sb} \sim 1.35$) as similar with the previously reported CSS- Sb_2Se_3 thin films (*Solar Energy Mater. Solar Cells*, **2018**, 188, 177). We also have measured the grain size distribution depending on the cooling rate as shown in Figure R2. The fast cooling sample revealed a slightly larger average value of grain size (~ 1038 nm) compared with the one of the slow cooling (~ 850 nm), while both samples had similar standard deviation (~ 370 nm). In our revised manuscript, we have added the results with relevant descriptions. We thank the reviewer again for the helpful comment.

Figure R1. Cooling rate comparison during close space sublimation process between the fast and slow cooling Sb_2Se_3 .

Figure R2. Grain size distribution of Sb_2Se_3 thin films depending on the cooling rate.

Revision made (colored blue):

(Line 11, Page 7)

We denoted the sample prepared with N₂-assisted cooling as the ‘fast-cooling’ sample, while the naturally cooled sample was denoted as the ‘slow-cooling’ sample. The fast cooling sample revealed a slightly larger average value of grain size (~1038 nm) compared with the one of the slow cooling (~850 nm), while both samples had similar standard deviation (~370 nm, Supplementary Fig. 3). The cooling rates for the fast cooling and the slow cooling until it has reached 200 °C are approximately 15.7 °C/min and 11.3 °C/min, respectively. In addition, the energy-dispersive X-ray spectroscopy (EDX) analysis showed that both fast and slow cooling Sb₂Se₃ films are slightly selenium poor (Se/Sb ~ 1.35) as similar with the previously reported CSS-Sb₂Se₃ thin films³¹.

Supplementary Fig. 3 | Grain size distribution of Sb₂Se₃ thin films depending on the cooling rate. a, Fast cooling Sb₂Se₃ and b, slow cooling Sb₂Se₃

Referee #1’s Comment 2:

It is surprising that the IPCE improved from 45% to 95% in the short wavelength after fast cooling (Fig. 2d). Is there any optical transmittance for the substrate, Au coated FTO substrate. How the Au deposited on the FTO and how thick the Au layer? As mentioned by the authors, the optical properties for the fast and slow cooling samples are nearly identical. As the

authors proposed, TiO₂ layer may contribute to the improved performance, however, the CSS can directly grow dense Sb₂Se₃ film (may cite Sol. RRL, 2: 1800128.2018 and Nano Energy, 49, 346, 2018). It is still unclear why the TiO₂ could boost the device performance significantly. In addition, in the KPFM images, the Fig.3c and 3f shown the surface profile. The KPFM tip may not record the 1 um valley (till the FTO substrate?)

Author's response:

The performance difference between the fast and slow cooling samples is due to the pin-hole formation and the resulting carrier recombination, as demonstrated by KPFM analysis. The enhancement by the fast cooling strategy is NOT related to the Au or TiO₂ layers as both of the fast and slow cooling samples have the same device configuration (FTO/Au/Sb₂Se₃/TiO₂/RuO_x, Fig. 2). We deposited 70 nm Au layer by a thermal evaporator. The role of the Au layer on Sb₂Se₃ photocathode was well investigated in our previous work (*ACS Energy Lett.*, **2019**, 4, 995). Briefly, the Au layer mainly acts as a hole selective contact, which facilitates the transfer of photo-generated holes. Of course, the 70 nm Au layer can reflect the incident light to some extent, but the more important role is selective hole transfer as proven by the fact that a transparent hole selective layer (Cu:NiO) is more efficient than reflective Au layer, indicating that the selective hole transport ability is more important than light reflection (*ACS Energy Lett.*, **2019**, 4, 995). The Au layer as a hole selective contact was also discussed in Sb₂Se₃ solar cells (*Solar Energy*, **2019**, 182, 96). As the reviewer pointed out, Sb₂Se₃ can be directly grown on FTO substrate without Au or TiO₂. We have taken some SEM images and measured the PEC performance without Au layer (please refer to the response to the comment #3 below). Regarding the role of TiO₂, it is well known that TiO₂ can act as a protective layer as well as an n-type semiconductor on a p-type semiconductor layer to form a p-n junction (*ACS Energy Lett.*, **2016**, 1, 1127; *J. Mater. Chem. A*, **2017**, 5, 2180; *ACS Energy Lett.* **2019**, 4, 517). Additionally, the AFM and KPFM techniques are widely used to determine surface topography and surface potential distribution. The height recorded by AFM and KPFM ranged from a few nm to hundreds nm and even sometimes μm scale (*Sol. Energy Mater. Sol. Cells*, **2018**, 183, 34; *Nano Converg.*, **2014**, 1, 27). In our data (Fig 3), it is also evident that a distinct potential peak was observed when there was a rapid drop of the height. In our revised manuscript, we have added some relevant references for explaining the capability of KPFM to determine a potential peak at a deep valley as follows.

Revision made (colored blue):

(Page 13, Line 7)

In contrast, in the slow-cooling sample, the surface potential increased significantly with a rapid drop in the topography (Fig. 3f), indicating direct contact between the n-type TiO₂ layer and substrate due to pin-holes. It might be worth to note that the height recorded by AFM and KPFM ranged from a few nm to hundreds nm and even sometimes μm scale³⁸⁻³⁹. In such a case, the photo-excited electrons can be extracted laterally to the ribbons and they can recombine with the holes at the back contact as shown in Fig. 3h due to the large electric field across the p–n junction.

38 Vishwakarma, M., Varandani, D., Andres, C., Romanyuk, Y. E., Haass, S. G., Tiwari, A. N., Mehta, M. R. A direct measurement of higher photovoltage at grain boundaries in CdS/ CZTSe solar cells using KPFM technique. *Sol. Energy Mater. Sol. Cells* **183**, 43-40 (2018)

39 Kim, G. Y., Kim, J., Jo, W., Son, D.-H., Kim, D.-H., Kang, J.-K. Nanoscale investigation of surface potential distribution of Cu₂ZnSn(S,Se)₄ thin films grown with additional NaF layers. *Nano Convergence* **1**, 27 (2014)

Referee #1's Comment 3:

Again, the role of the Au layer on the FTO on the PEC performance is unclear, it should compare the sample with and without Au layer. Here, the Sb₂Se₃ grown on the FTO.

Author's response:

As we mentioned in the above comments, the role of the Au layer as a hole selective contact was elucidated in our previous study (*ACS Energy Lett.*, **2019**, 4, 995), as well as in Sb₂Se₃ solar cell research (*Solar Energy*, **2019**, 182, 96). Here, as per the reviewer's comment, we have compared the PEC performance and microstructures of Sb₂Se₃ photocathodes depending on the presence/absence of the Au layer. As shown in Figure R3, there was no noticeable microstructural difference between Sb₂Se₃ thin films with and without the Au layer. On the

other hand, PEC performance of Sb_2Se_3 photocathodes significantly increased with the Au layer as shown in Figure R4. The results are in accordance with the previous studies, which showed much enhanced performance with the Au hole selective contact (*ACS Energy Lett.*, **2019**, *4*, 995; *ACS Nano*, **2018**, *12*, 11088). In our revised manuscript, we have added the results of without Au layer sample with relevant descriptions and references. We thank the reviewer for the helpful comment improving the quality of our manuscript.

Figure R3. SEM images of Sb_2Se_3 (a-b) without and (c-d) with the Au bottom contact layer.

Figure R4. PEC performance of (a) RuO_x/TiO₂/Sb₂Se₃/FTO and (b) RuO_x/TiO₂/Sb₂Se₃/Au/FTO photocathodes.

Revision made (colored blue):

(Page 10, Line 2)

Fig. 2a–b show the PEC performance of RuO_x/TiO₂/Sb₂Se₃/Au/FTO photocathodes using fast-and slow-cooling Sb₂Se₃ films measured in pH 1 electrolytes. As we mentioned above, the Au layer acts as a hole selective contact, which facilitates the transfer of photo-generated holes while blocking the electrons backflow³²⁻³³. Without the Au layer, Sb₂Se₃ photocathodes revealed relatively poor performance while nearly similar morphology of Sb₂Se₃ was observed (Supplementary Fig. 4), which verifies the role of the Au layer not affecting the growth of Sb₂Se₃, but assisting the transfer of photo-generated charges. The RuO_x catalytic layer was deposited by the PEC method, while atomic layer deposition (ALD) was used for the TiO₂ layer, similar to a previous study³². In both samples, the onset potentials shifted towards a positive direction after the first scan due to activation of the RuO_x catalyst³⁴.

(Supplementary information)

Supplementary Fig. 4 | Sb₂Se₃ photocathodes without the Au bottom contact layer. a-b, SEM images of fast-cooling Sb₂Se₃ on FTO substrate. The morphology of Sb₂Se₃ directly grown on the FTO substrate is nearly similar to that of Sb₂Se₃ grown on Au/FTO. **c,** J-V curves of RuO_x/TiO₂/Sb₂Se₃/FTO photocathodes in pH 1 H₂SO₄ electrolytes.

Referee #1's Comment 4:

How the cells chemical composition post reliability test?

Author's response:

We have measured Raman spectroscopy to investigate the chemical composition difference after the reliability test as per the reviewer's comment. Before the stability test, the Raman spectra of Sb_2Se_3 shows one distinct peak at $\approx 190 \text{ cm}^{-1}$ along with a shoulder peak at $\approx 208 \text{ cm}^{-1}$, both of which are attributed to the vibration modes in Sb_2Se_3 phase (Figure R5a). After the stability test, an additional peak located at $\approx 250 \text{ cm}^{-1}$ appeared. The additional peak indicates the formation of by-products such as Sb_2O_3 ($\approx 254 \text{ cm}^{-1}$) and/or several Se phases (e.g., monoclinic Se_8 rings at $\approx 253 \text{ cm}^{-1}$, rhombohedral Se_6 rings at $\approx 247 \text{ cm}^{-1}$, and amorphous Se at $\approx 250 \text{ cm}^{-1}$), which result from the decomposition of Sb_2Se_3 . In addition, there was also morphological destruction after the stability test (Figure R5b-c). According to our previous study on the stability of Sb_2Se_3 photocathodes (*Adv. Energy Mater.*, **2019**, *9*, 1900179), the morphological destruction of Sb_2Se_3 photocathode is caused by the photo-reduction of TiO_2 accompanied by the degradation of Sb_2Se_3 . In the revised manuscript, we have added the chemical composition results with relevant descriptions. We thank the reviewer for the helpful comment.

Figure R5. Chemical composition and microstructures of Sb_2Se_3 before and after stability test. (a) Raman spectra of Sb_2Se_3 photocathodes and SEM images of (b) before and (c) after stability test.

Revision made (colored blue):

(Page 15, Line 9)

The $\text{RuO}_x/\text{TiO}_2/\text{Sb}_2\text{Se}_3$ sample retained approximately 60% of initial photocurrent density after 35 hours in the neutral electrolytes, which is the best stability of Sb_2Se_3 photocathodes reported so far (Supplementary Fig. 9a). The photocurrent density of $\text{Pt}/\text{TiO}_2/\text{Sb}_2\text{Se}_3$ decreased more rapidly (42% photocurrent after 5 h), probably due to the larger bubbles at Pt surfaces, as evidenced by severe fluctuations in the enlarged photocurrent curves

(Supplementary Fig. 9b–c). We measured Raman spectroscopy to investigate the chemical composition variation after the reliability test. Before the stability test, the Raman spectra of the RuO_x/TiO₂/Sb₂Se₃/Au/FTO photocathode showed one distinct peak at $\approx 190\text{ cm}^{-1}$ along with a shoulder peak at $\approx 208\text{ cm}^{-1}$, both of which are attributed to the vibration modes in Sb₂Se₃ phase (Supplementary Fig. 8a). After the stability test, an additional peak located at $\approx 250\text{ cm}^{-1}$ appeared. The additional peak indicates the formation of by-products such as Sb₂O₃ ($\approx 254\text{ cm}^{-1}$) and/or several Se phases (e.g., Se₈ rings at $\approx 253\text{ cm}^{-1}$, Se₆ rings at $\approx 247\text{ cm}^{-1}$, and amorphous Se at $\approx 250\text{ cm}^{-1}$) as a result from the decomposition of Sb₂Se₃. In addition, there was also morphological destruction after the stability test (Supplementary Fig. 8b-c). According to our previous study on the stability of Sb₂Se₃ photocathodes²⁵, the morphological destruction of Sb₂Se₃ photocathode is caused by the photo-reduction of TiO₂ accompanied by the degradation of Sb₂Se₃.

(Supplementary Information)

Supplementary Fig. 8 | Chemical composition and microstructures of Sb₂Se₃ before and after stability test. a, Raman spectra of Sb₂Se₃ photocathodes and SEM images of **b**, before and **c**, after stability test.

Referee #2

Comments to the Author

The manuscript by Yang et al. describes a new synthesis method for Sb₂Se₃ photocathodes—a highly promising material for practical water splitting—and the development of a tandem cell with BiVO₄. The performance of the Sb₂Se₃ is outstanding, the article is well written and appropriately cited, and the level of scientific discussion is high. I recommend to accept after addressing the following minor concerns:

Author's response:

We thank the reviewer for evaluating our work as a well-written and meaningful work. All of the comments made by the reviewer are helpful for improving the overall quality of our work. Our detailed, point-by-point responses to the reviewer's comments can be found below.

Referee #2's Comment 1:

The title should be more descriptive of the contents of the article.

Author's response:

We have modified the title as per the reviewer's comment as follows.

Revision made (colored blue):

Benchmark performance of low-cost Sb_2Se_3 photocathodes obtained by the fast-cooling strategy during close space sublimation for unassisted solar overall water splitting

Referee #2's Comment 2:

Page 15: "The photocurrent density of Pt/TiO₂/Sb₂Se₃ decreased more rapidly (42% photocurrent after 5 h), probably due to the larger bubbles at Pt surfaces, as evidenced by severe fluctuations in the enlarged photocurrent curves (Supplementary Fig. 7b–c)." what do the authors mean, that the releasing of large bubbles cause the Pt to detach?

Author's response:

The detachment of Pt particles due to the releasing of large bubbles is one of the well-known degradation mechanisms in the Pt-decorated photocathodes for water splitting. It is widely reported in the literature (please refer to the section 4.3.2 in *Chem. Soc. Rev.*, **2019**, 48, 4979). In the revised manuscript, we have added relevant descriptions and references on the detachment of Pt for better understanding.

Revision made (colored blue):

(Page 15, Line 12)

The photocurrent density of Pt/TiO₂/Sb₂Se₃ decreased more rapidly (42% photocurrent after 5 h), probably due to the larger bubbles at Pt surfaces, as evidenced by severe fluctuations in the enlarged photocurrent curves (Supplementary Fig. 7b–c). It should be noted that the

detachment of Pt particles due to the releasing of large bubbles is one of the well-known degradation mechanisms in the Pt-decorated photocathodes for water splitting⁴².

(Reference)

42 Yang, W., Prabhakar, R. R., Tan, J., Tilley, S. D., Moon, J., Strategies for enhancing the photocurrent, photovoltage, and stability of photoelectrodes for photoelectrochemical water splitting. *Chem. Soc. Rev.*, **48**, 4979 (2019).

Referee #2's Comment 3:

Page 18: “However, at a high concentration of 1.0 M, the stability of the photocathode degraded” Any thoughts why?

Author's response:

It is known that the photocurrent density of photocathodes can increase as the concentration of the electrolyte increases due to the reduced resistance of the electrolyte (*Energy Environ. Sci.*, **2018**, *11*, 3003-3009). The increased photocurrent density is indicative of the larger hydrogen bubbles, which could cause the instability of the photocathodes due to the detachment of Pt as mentioned above. In the revised manuscript, we have added the relevant descriptions and references for clarifying the results.

Revision made (colored blue):

(Page 18, Line 9)

However, at a high concentration of 1.0 M, the stability of the photocathode degraded, presumably due to the accelerated Pt detachment in the higher photocurrent condition.

Referee #2's Comment 4:

Page 20: “However, the photocurrent density of Sb₂Se₃ photocathodes at 0.4 VRHE with and without Sb₂Se₃ ...” with and without CdS?

Author's response:

It seems we made a mistake. The original meaning is “with and without BiVO₄”. We have corrected the typo in the revised manuscript. We thank the reviewer for the helpful comment.

Revision made (colored blue):

(Page 20, Line 22)

However, the photocurrent density of Sb_2Se_3 photocathodes at $0.4 V_{\text{RHE}}$ with and without BiVO_4 were 5.0 mA cm^{-2} and 2.2 mA cm^{-2} , respectively, which are higher than the photocurrents shown by the J–V curve.

Referee #2's Comment 5:

Fig 5 caption: “All analyses were conducted in 0.5 M phosphate buffer + 0.01 M V_2O_5 (pH 7.0)” – were faradaic efficiency measurements carried out to ensure that the 20 mM vanadium ions do not interfere with the hydrogen production? Since V^{5+} is much more easily reduced than proton.

Author's response:

We thank the reviewer's comment. As the reviewer pointed out, theoretically the V^{5+} can be reduced prior to proton, possibly affecting the performance of our Sb_2Se_3 photocathode-based tandem devices for water splitting. However, some previous studies showed that the presence of V^{5+} in a potassium borate buffer solution doesn't produce any additional reduction or oxidation peaks and doesn't interfere with water reduction and water oxidation (*Nature Energy*, **2018**, 3, 53). In order to verify whether V^{5+} participates in the redox reactions or not, we have performed additional experiments as follows. As shown in Figure R5a, there are distinctive peaks in the LSV scans for a Pt electrode upon addition of V^{5+} into strongly acidic electrolyte, indicative of a significant reduction of V^{5+} . In contrast, there is no noticeable difference between with/without V^{5+} electrolyte when measured in a neutral electrolyte (0.5 M KPi). These results imply that the reactivity of V^{5+} , which is relatively stronger in an acidic electrolyte, significantly decreases in a neutral electrolyte. As we measured our Sb_2Se_3 -based tandem device in a neutral electrolyte (0.5 M KPi), there is no significant change of both the Sb_2Se_3 photocathode and the BiVO_4 photoanode upon adding V^{5+} into our electrolyte as shown in Figure R6. It should be noted that the slight difference observed in the photocathode case (Figure R6d), possibly due to parasitic light absorption by yellow V^{5+} ions, does not affect the performance of our tandem device as the operation potential of the tandem device is around $0.4 V_{\text{RHE}}$. **Accordingly, in any cases, it is reasonable to conclude that addition of V^{5+} does not interfere with the hydrogen production by our Sb_2Se_3 -based tandem device.** We have modified our manuscript with the results and descriptions on the reduction of V^{5+} . We appreciate the reviewer for improving the quality of our work.

Figure R5. Linear sweep voltammogram of Pt rod with/without V^{5+} in (a) a strongly acidic electrolyte and (b) a neutral electrolyte.

Figure R6. (a) Photograph of vanadium oxide-dissolved phosphate buffer. A slightly yellow hue was observed. (b) Stability of two $BiVO_4$ - Sb_2Se_3 tandem cells in different electrolytes at

a constant potential (0.0 V against the counter electrode). (c-d) J–V curves for each photoelectrode without or with dissolved vanadium oxide.

Revision made (colored blue):

(Page 19, Line 6)

However, owing to the low stability of BiVO₄ in phosphate, fast degradation of performance was observed for the tandem cell, and we addressed the stability issue by adding vanadium cation (V⁵⁺) as done by Choi group⁴⁵. It should be noted that theoretically the V⁵⁺ can be reduced prior to proton, possibly affecting the performance of our Sb₂Se₃ photocathode-based tandem devices for water splitting. As shown in Supplementary Fig. 16a, there are distinctive peaks in the LSV scans for a Pt electrode upon addition of V⁵⁺ into strongly acidic electrolyte, indicative of a significant reduction of V⁵⁺. In contrast, there is no noticeable difference between with/without V⁵⁺ electrolyte when measured in a neutral electrolyte (0.5 M KPi, Supplementary Fig. 16b). These results imply that the reactivity of V⁵⁺, which is relatively stronger in an acidic electrolyte, significantly decreases in a neutral electrolyte. As we measured our Sb₂Se₃-based tandem device in a neutral electrolyte (0.5 M KPi), there is no significant change of both the Sb₂Se₃ photocathode and the BiVO₄ photoanode upon adding V⁵⁺ into our electrolyte as shown in Supplementary Fig. 17. It is also noteworthy that the slight difference observed in the photocathode case (Supplementary Fig. 17d), possibly due to parasitic light absorption by yellow V⁵⁺ ions, does not affect the performance of our tandem device as the operation potential of the tandem device is around 0.4 V_{RHE}. Accordingly, in any cases, it is reasonable to conclude that addition of V⁵⁺ does not interfere with the hydrogen production by our Sb₂Se₃-based tandem device.

(Supplementary Information)

Supplementary Fig. 16 | Linear sweep voltammogram of Pt rod with/without V^{5+} in **a**, a strongly acidic electrolyte and **b**, a neutral electrolyte.

Supplementary Fig. 17 | **a**, Photograph of vanadium oxide-dissolved phosphate buffer. A slightly yellow hue was observed. **b**, Stability of two $BiVO_4$ - Sb_2Se_3 tandem cells in different electrolytes at a constant potential (0.0 V against the counter electrode). **c-d**, J-V curves for each photoelectrode without or with dissolved vanadium oxide.

Referee #2's Comment 6:

Page 29: “and simulated solar light illumination (AM 1.5G, Newport Corporation) was used as the light source.” How was the intensity calibrated?

Author's response:

Calibration of the 1-sun level was performed using a standard Si reference cell certified by the Newport Corporation, consisting of a readout device and a $2 \times 2 \text{ cm}^2$ calibrated solar cell made of monocrystalline silicon. During calibration, the Si reference cell was located at the same position of the sample for PEC measurement. We have added the calibration process in the revised manuscript.

Revision made (colored blue):

(Methods)

The Sb_2Se_3 photocathodes were submerged in an acidic (H_2SO_4 , pH ~ 1) or a neutral (phosphate buffer, pH ~ 6.25) electrolyte, and simulated solar light illumination. Calibration of the 1-sun level was performed using a standard Si reference cell certified by the Newport Corporation, consisting of a readout device and a $2 \times 2 \text{ cm}^2$ calibrated solar cell made of monocrystalline silicon. During calibration, the Si reference cell was located at the same position of the sample for PEC measurement.

Referee #3**Comments to the Author**

The paper, “Benchmark performance of earth-abundant Sb_2Se_3 photocathodes for unassisted solar overall water splitting by Yang et al. presents a tandem Pt/TiO₂/CdS/ Sb_2Se_3 /Au/FTO photocathode and NiFeOx/BiVO₄/FTO photoanode that without a bias shows $\sim 1 \text{ mA/cm}^2$ current density with $\sim 1.1\%$ STH using simulated AM1.5G 100 mW/cm² light.

There are no major concerns regarding the data or how the authors have conducted the experiments (although there are a couple of minor questions outlined below). The work is well done. However, there is little new science brought forth. Sb_2Se_3 modified with CdS and TiO₂ with at Pt catalyst has already been reported as a photocathode (see ACS Nano 2017, 11,

12753). The 1.1% sustained STH efficiency over 10 h is neither first nor best, and remains too low for practical water splitting. Admittedly the anode is the bigger problem, but it's not clear why this paper should be published in Nature Communications when many of the other references in Supplementary Table S1 are in a different tier of journals. This paper provides a few insights into materials engineering and device considerations, but even then, the ideas aren't new. Consequently, this paper is not recommended for publication in Nature Communications.

Author's Response:

We thank the reviewer for stating that “There are no major concerns regarding the data or how the authors have conducted the experiments” and evaluating our manuscript as a well-done work. As the reviewer pointed out, there are many nice papers pertaining to photocathode materials in a different tier of journals (Supplementary Table S1). However, we would like to emphasize that our Sb₂Se₃-based PEC device has sufficient novelty to be published in Nature Communication in terms of not only the performance and but also the material's novelty. We are well aware that the development of cost-effective materials still remains a paramount challenge for the commercialization of PEC water splitting, despite the tremendous effort has devoted by researchers over decades. Sb₂Se₃ is one of the attractive emerging materials for PEC water splitting in terms of cost, band gap, optoelectronic properties, photocorrosion stability, and processability (please refer to the recent highlight paper “*Rapid advances in antimony triselenide photocathodes for solar hydrogen generation*”, *J. Mater. Chem. A*, **2019**, 7, 20467). Thus, establishing a new benchmark performance of this emerging photocathode could be much more important than other high performance devices based on well-investigated materials. For example, we modified Table S1 to emphasize the novelty of our system compared with other photocathode materials (for review purpose only). Additionally, we also report a novel strategy by using close-space sublimation, which is a relative scalable method. Our fast cooling strategy enabling smooth and pin-hole-free Sb₂Se₃ thin films provides meaningful insight into how thermodynamically metastable morphology can be achieved. We believe that our finding contributes to other materials systems as well as the development of materials science. Of course, the performance and stability of our Sb₂Se₃ based photocathodes should be further improved for practical water splitting. However, we believe that reporting the highest efficiency of the novel and emerging semiconductor can sufficiently provide the feasibility to be commercialized in the future as well as it is worth to

receive attention from researchers worldwide. Therefore, we politely ask the reviewer to re-consider recommending our manuscript which has been revised based on the reviewer's comment below.

*Ref.	Photocathode	***Efficiency	Ref	Remarks
(1)	Pt/TiO ₂ /CdS/Sb ₂ Se ₃ /Au/FTO	1.5%, stable for 10 h	This work	Emerging material (the first Sb ₂ Se ₃ photocathode reported in 2017) Low-cost No possible secondary phases (easy to synthesize) Theoretical max. STH > 20 % Good performance
(2)	TiNi/p-Si	Tandem scheme HC-STH: 0.05%	Lai 2015 ¹	No novelty on semiconductor (Si)
(3)	Pt/p-Si	Tandem scheme STH: 0.57%	Xu et al., 2016 ²	No novelty on semiconductor (Si)
(4)	Pt/TiO ₂ /a-Si	Tandem scheme STH: 0.91%, stable for 10 h	Jang et al., 2015 ³	No novelty on semiconductor (Si)
(5)	NiMo+SiO ₂ (partially coated)/n/p-Si nanowire	Tandem scheme STH: 2.1%, stable for 2 h	Vijselaar et al., 2018 ⁴	No novelty on semiconductor (Si)
(6)	Pt/TiO ₂ /Zn:InP	Parallel scheme STH: 0.5%	Kornienko et al., 2016 ⁵	Expensive material Complicated synthesis methods
(7)	Pt/In ₂ S ₃ /CdS/Cu ₂ ZnSnS ₄ Mo/SLG	Parallel scheme HC-STH: 0.28%	Jiang et al., 2015 ⁶	Possible secondary phases (SnS ₂ , ZnS, CuZnS ₂ ...) Low-performance
(8)	Pt/Mo/Ti/CdS/In ₂ S ₃ /(ZnSe) _{0.85} (CuIn _{0.7} Ga _{0.3} Se ₂) _{0.15} /Mo/SLG/Ti foil	Parallel scheme STH: 1.0%	Higashi et al., 2017 ⁷	Expensive elements (In, Ga) Possible secondary phases
(9)	Pt/Mo/Ti/(ZnSe) _{0.85} (CIGS) _{0.15} /Mo/SLG	Parallel scheme STH: 0.91%	Kaneko et al., 2016 ⁸	Expensive elements (In, Ga) Possible secondary phases
(10)	Pt/ZnS/CdS/(ZnSe) _{0.85} (CuIn _{0.7} Ga _{0.3} Se ₂) _{0.15} /Mo/SLG	Parallel scheme STH: 0.6%	Goto et al., 2017 ⁹	Expensive elements (In, Ga) Possible secondary phases
(11)	Pt/CdS/CuGa ₃ Se ₅ /(Ag, Cu)GaSe ₂ /Mo/SLG	Tandem scheme STH: 0.67%	Kim et al., 2016 ¹⁰	Expensive elements (Ga, Ag) Possible secondary phases

		stable for 2 h		
(12)	Pt/TiO ₂ /CdS/(CuGa _{1-y} In _y) _{1-x} Zn _{2x} S ₂ /Au	Parallel scheme STH: 1.1%	Hayashi et al., 2018 ¹¹	Expensive elements (In, Ga) Possible secondary phases
(13)	Pt/HfO ₂ /CdS/Cu ₂ ZnSnS ₄ Mo/SLG	STH: 1.046%, stable for 10 h	Huang et al., 2018 ¹²	Possible secondary phases (SnS ₂ , ZnS, CuZnS ₂ ..)
(14)	Pt/TiO ₂ /Al ₂ O ₃ /CdS/CIGS/Mo/SLG	STH: 1.01%, stable for >0.5 h	Chen et al., 2018 ¹³	Expensive elements (In, Ga) Possible secondary phases
(15)	Pt/TiO ₂ /CdS/CuIn _{0.5} Ga _{0.5} Se ₂ /Mo/SLG	STH: 3.7%, stable for >0.5 h	Kobayashi et al., 2018 ¹⁴	Expensive elements (In, Ga) Possible secondary phases
(16)	Pt/In ₂ S ₃ /CdS/(ZnSe) _{0.85} (CuIn _{0.7} Ga _{0.3} Se ₂) _{0.15} /Mo/SLG	STH: 1.0%, stable for 50 h (60% retained)	Kaneko et al., 2018 ¹⁵	Expensive elements (In, Ga) Possible secondary phases
(17)	RuO ₂ /TiO ₂ /AZO/Cu ₂ O/Au/FTO	Tandem scheme STH: 1.0% (retained less than 5 min)	Bornoz et al., 2014 ¹⁶	Vulnerable to corrosion Low theoretical Max. STH (~10 %)
(18)	RuO ₂ /TiO ₂ /Ga ₂ O ₃ /Cu ₂ O/ Au/FTO	STH: 3.0%, stable for 12 h (90% retained)	Pan et al., 2018 ¹⁷	Vulnerable to corrosion Low theoretical Max. STH (~10 %)
(19)	Pt/CuBi ₂ O ₄ /FTO	STH: 0.15% Unstable	Kim et al., 2018 ¹⁸	Low-performance Possible secondary phases
(20)	Pt/Ag/PEIE/PCBM/(CsFAMA)Pb I ₃ /NiO/FTO	STH: 0.59%, stable for 18 h	Andrei et al., 2018 ¹⁹	Low-performance Vulnerable to corrosion

Referee #3's Comment 1:

In the XRD shown in Figure 1, what would be the relative intensities of the (221) and (301) Bragg reflections in a truly randomly oriented sample? Is that what is shown in JCPDS 15-0861? The data for the fast and slow cooling look similar enough that I'm not convinced regarding the statement that a slight change in the XRD data implies that ribbons move during cooling.

Author's response:

To quantify the relative intensities of each plane revealed in XRD data to a standard Sb₂Se₃ powder (JCPDS 15-8601), we have calculated the texture coefficient T_c, which is defined as

$$T_c(hkl) = n \frac{I(hkl)/I_o(hkl)}{\sum_j^n I(hkl)/I_o(hkl)}$$

where $I(hkl)$ is the measured relative intensity of the peak corresponding to the hkl diffraction, $I_o(hkl)$ is the relative intensity from a standard powder sample (JCPDS 15-0861), and n is the total number of diffraction peaks used in the evaluation. A large T_c value for a specific diffraction peak indicates preferred orientation along this direction. In the present case, we chose four diffraction peaks ($n = 4$) corresponding to 2θ values of 120, 211, 221, and 301. Figure R7 clearly shows that T_c (120) of both fast and slow cooling samples is nearly zero while the other values are higher than 1, indicating both samples have (hkl) preferred

orientation. Although both samples have a similar preferred orientation, it is also obvious that the fast cooling sample revealed higher T_c values of (211) and (301) planes and lower T_c value of (221) plane, implying possible rearrangement of the ribbons. In our revised manuscript, we have added the quantitative analysis of the ribbon orientations based on the T_c values with relevant descriptions. We thank the reviewer for improving the quality of our work.

Figure R7. The texture coefficients of selected diffraction peaks in different Sb_2Se_3 thin films.

Revision made (colored blue):

(Page 7, Line 15)

As found in previous studies on Sb_2Se_3 thin-film solar cells, (hk1) orientations, representing $(Sb_4Se_6)_n$ nanoribbons oriented perpendicular or inclined relative to the substrate (Fig. 1h, for example), are advantageous for a superior performance owing to efficient carrier transport along the [001] direction^{18,19}. To quantify the relative intensities of each plane revealed in XRD data to a standard Sb_2Se_3 powder (JCPDS 15-8601), we have calculated the texture coefficient T_c , which is defined as:

$$T_c(hkl) = n \frac{I(hkl)/I_o(hkl)}{\sum_1^n I(hkl)/I_o(hkl)}$$

where $I(hkl)$ is the measured relative intensity of the peak corresponding to the hkl diffraction, $I_o(hkl)$ is the relative intensity from a standard powder sample (JCPDS 15-0861), and n is the total number of diffraction peaks used in the evaluation. A large T_c value for a specific diffraction peak indicates preferred orientation along this direction. In the present case, we chose four diffraction peaks ($n = 4$) corresponding to 2θ values of 120, 211, 221, and 301. Supplementary Fig. 4 clearly shows that T_c (120) of both fast and slow cooling samples is nearly zero while the other values are higher than 1, indicating both samples have (hkl) preferred orientation. Although both samples have a similar preferred orientation, it is also obvious that the fast cooling sample revealed higher T_c values of (211) and (301) planes and lower T_c value of (221) plane, implying possible rearrangement of the ribbons.

(Supplementary Information)

Supplementary Fig. 4 | The texture coefficients of selected diffraction peaks in different Sb_2Se_3 thin films.

Referee #3's Comment 2:

In Figure 2a-b, the origin of the shift to more positive potentials after the 1st sweep is

described as RuO_x activation? Is the surface the same after this activation? What do the SEM images look like after HER?

Author's response:

As the reviewer mentioned, it is correct that the origin of the shift to more positive potentials after the 1st sweep is described as RuO_x activation. As shown in Figure R8a-b, it seems that the particle size of RuO_x slightly decreases upon activation, the degree of the change is not significant before and after activation. Additionally, there was some morphological destruction after the stability test (Figure R8b-c). According to our previous study on the stability of Sb₂Se₃ photocathodes (*Adv. Energy Mater.*, **2019**, 9, 1900179), the morphological destruction of Sb₂Se₃ photocathode is caused by the photo-reduction of TiO₂ accompanied by the degradation of Sb₂Se₃. In the revised manuscript, we have added some descriptions on the changes after the stability test.

Figure R8. SEM images of RuO_x/TiO₂/Sb₂Se₃/Au/FTO photocathodes (a) before and (b) after the activation of the RuO_x layer.

Revision made (colored blue):

(Page 15, Line 9)

In addition, there was also morphological destruction after the stability test (Supplementary Fig. 8b-c). According to our previous study on the stability of Sb₂Se₃ photocathodes²⁵, the morphological destruction of Sb₂Se₃ photocathode is caused by the photo-reduction of TiO₂ accompanied by the degradation of Sb₂Se₃.

(Supplementary Information)

Supplementary Fig. 5 | Chemical composition and microstructures of Sb₂Se₃ before and after stability test. **a**, Raman spectra of Sb₂Se₃ photocathodes and SEM images of **b**, before and **c**, after stability test.

Referee #3's Comment 3:

Can the authors avoid mixing thermodynamic and kinetic language? Current is cathodic (or anodic), but there is no such thing as cathodic potential. So, rather than scanning in the “cathodic direction,” the authors should state that J-V curves were scanned from open circuit to more negative potential.

Author's response:

In the revised manuscript, we have modified the confusing descriptions as per the reviewer's comment. We thank the reviewer for clarifying our work.

Revision made (colored blue):

(Figure 2)

Fig. 2. | PEC performance of RuO_x/TiO₂/Sb₂Se₃/Au/FTO photocathodes in pH 1 H₂SO₄ electrolyte. **a-b**, J–V curves of (a) fast- and (b) slow-cooling samples under simulated 1 sun air mass 1.5 G chopped illumination at a scan speed of 5 mV s⁻¹ from positive to negative potential in the cathodic direction and **c**, corresponding HC-STH efficiencies. **d**, Wavelength-dependent IPCE and integrated photocurrent density of fast- and slow-cooling samples measured at 0 V_{RHE}.

(Figure 4)

Fig. 4. | Microstructure and PEC performance of Pt/TiO₂/CdS/Sb₂Se₃/Au/FTO photocathodes. **a**, Cross-sectional SEM image of TiO₂/CdS/Sb₂Se₃ on Au/FTO substrate. **b**,

J–V curves for Sb_2Se_3 photocathodes with/without a CdS layer in pH 1 H_2SO_4 under simulated 1 sun air mass 1.5 G chopped illumination at a scan speed of 5 mV s^{-1} from positive to negative potential ~~in the cathodic direction~~ and **c**, corresponding HC-STH efficiencies. **d**, Wavelength-dependent IPCE at 0 V_{RHE} of Sb_2Se_3 photocathodes with/without CdS layer.

Referee #3's Comment 4:

Can the authors provide data for multiple films? It is not clear if the data are representative of typical films or the best-performing films.

Author's response:

We thank the reviewer for the careful comment. Normally we presented the best-performing films. For example, for $\text{RuO}_x/\text{TiO}_2/\text{Sb}_2\text{Se}_3/\text{Au}/\text{FTO}$ photocathodes, which we used as a benchmark photocurrent density, 30 mA cm^{-2} is the highest value, while normally $25 - 30 \text{ mA cm}^{-2}$ photocurrent is observed as shown in Figure R9. In addition, for the tandem devices, For tandem cell, photocurrent density near $0.4 V_{\text{RHE}}$ of each photoanode and photocathode determines the overall STH efficiency. As shown in Figure R10, photocurrent density of BiVO_4 at $0.4 V_{\text{RHE}}$ varies from 1.2 to 0.8 mA cm^{-2} , thus the overall STH efficiencies of the tandem cell range from 1.48 % to 0.98 %. We have modified the descriptions regarding benchmark performance in our revised manuscript.

Figure R9. J-V curves of six different $\text{RuO}_x/\text{TiO}_2/\text{Sb}_2\text{Se}_3/\text{Au}/\text{FTO}$ photocathodes in pH 1 H_2SO_4 under simulated 1 sun air mass 1.5 G chopped illumination at a scan speed of 5 mV s^{-1} from positive to negative potential.

Figure R10. J-V curves of three different $\text{NiFeO}_x/\text{H},\text{Mo}:\text{BiVO}_4$ photoanodes (a) in the 0.5 M KPi and (b) 0.5 M KPi + 0.01 M V_2O_5 electrolyte.

Revision made (colored blue):

(Page 10, Line 6)

The photocurrent density of the fast-cooling sample approached 30 mA cm^{-2} at 0 V_{RHE} , which is not only the highest value obtained for a Sb_2Se_3 photocathode but also among the best observed for all photoelectrodes used in PEC water splitting so far. Note that the data shown in Fig.2 were obtained from the best performing device, while normally $25 - 30 \text{ mA cm}^{-2}$

at 0 V_{RHE} photocurrent density was observed in the fast-cooling Sb₂Se₃ based photocathodes.

(Page 19, Line 9)

As shown in Fig. 5b, the operating point of the two photoelectrodes, as estimated by the intersection of two J–V curves, was approximately 1.2 mA cm⁻² at 0.4 V_{RHE}, which corresponded to a STH efficiency of 1.5%. It should be noted that photocurrent density of BiVO₄ at 0.4 V_{RHE} varies from 1.2 to 0.8 mA cm⁻², thus the overall STH efficiencies of the tandem cell range from 1.48 % to 0.98 %.

Editorial comments

In an effort to ensure reproducibility of research data, we now also require that you provide a separate source data file. The source data file should, as a minimum, contain the raw data underlying all reported averages in graphs and charts, and uncropped versions of any gels or blots presented in the figures. To learn more about our motivation behind this policy, please see <https://www.nature.com/articles/s41467-018-06012-8>.

Within the source data file, each figure or table (in the main manuscript and in the Supplementary Information) containing relevant data should be represented by a single sheet in an Excel document, or a single .txt file or other file type in a zipped folder. Blot and gel images should be pasted in and labelled with the relevant panel and identifying information such as the antibody used. We also encourage you to include any other types of raw data that may be appropriate. An example source data file is available demonstrating the correct format: <https://www.nature.com/documents/ncomms-example-source-data.xlsx>

The file should be labelled ‘Source Data’, with the title and a brief description included in your cover letter, and should be mentioned in all relevant figure legends using the template text below:

“Source data are provided as a Source Data file.”

Response:

As per the editor’s comment, we have provided the source data in excel format and mentioned in all figure legends.

Reviewers' comments:

Reviewer #1 (Remarks to the Author):

The authors replied well to the reviewers' comments. As shown in the comments of Reviewer #3 and the authors' response, the work seems promising and demonstrates a benchmark photocurrent performance for Sb₂Se₃ photocathodes deposited using the CSS system. The reviewer agrees with the authors that Sb₂Se₃ is an emerging candidate for the photoelectrochemical application, but the authors reported lots of previous similar work, such as ACS Energy Lett., 2019, 4, 995; ACS Nano, 2018, 12, 11088, with similar device structure. Here, the CSS deposition of the Sb₂Se₃ absorber layer for photovoltaics application is not a novelty (Solar Energy Mater. Solar Cells, 2018, 188, 177, Sol. RRL, 2: 1800128.2018 and Nano Energy, 49, 346, 2018) although the authors claim the cooling rate could significantly impact on the photoelectrochemical performance. However, the fundamental mechanism for improved PEC performance is not clear, particularly, the chemical composition, microstructure did not show significant differences.

Particularly, the mechanism for the Au layer increase almost 3 times of the photocurrent (Fig.R4) has no detailed analysis, but only cite the previous work. And the low overpotential of the Sb₂Se₃, i.e., 0.4V still lower than other systems has not been investigated. By considering the Pt or RuOx cocatalyst and the 70 nm Au hole collection layer are all precious materials in the RuOx/TiO₂/Sb₂Se₃/Au/FTO and Pt/TiO₂/CdS/Sb₂Se₃/Au/FTO structure will significantly limit the scale application and may weaken the claimed "low cost" Sb₂Se₃ but an expensive device. By forming a tandem PEC cells with BiVO₄, the efficiency is not promising for 10 h stability. Overall, it does not show clearly innovative evidence and understanding to publish in Nature Communications when more similar results are published in other journals. Thus, I recommend the manuscript to be rejected and may submit to elsewhere.

Reviewer #2 (Remarks to the Author):

The authors have carefully considered and thoroughly and satisfactorily responded to all of my concerns, and the manuscript is now suitable for publication.

Reviewer #3 (Remarks to the Author):

The authors have suitably addressed all of the criticisms by all reviewers. The authors are commended for their diligence in conducting new experiments and including new analysis to answer each question. This reviewer is convinced that the findings are sufficiently distinct and novel to warrant publication in *Nature Communications*. In particular, carefully probing the optoelectronic properties of the multi-junction device exposes relevant questions and limitations for each component, which adds to new materials chemistry. Generating pinhole-free Sb₂Se₃ is crucial for long-term overall solar water splitting at zero bias.

Response Letter

Title: “Benchmark performance of low-cost Sb_2Se_3 photocathodes obtained by the fast-cooling strategy during close space sublimation for unassisted solar overall water splitting”

Reviewer #1

General comment:

The authors replied well to the reviewers' comments. As shown in the comments of Reviewer #3 and the authors' response, the work seems promising and demonstrates a benchmark photocurrent performance for Sb_2Se_3 photocathodes deposited using the CSS system.

General reply:

We thank the reviewer for mentioning that our work seems promising and demonstrates a benchmark performance for Sb_2Se_3 photocathodes, which is an emerging low-cost material. While we understand the reviewer's concern in some points, however, we respectfully disagree with the reviewer for the other points. Detailed point-by-point response can be found below.

Comment #1:

The reviewer agrees with the authors that Sb_2Se_3 is an emerging candidate for the photoelectrochemical application, but the authors reported lots of previous similar work, such as ACS Energy Lett., 2019, 4, 995; ACS Nano, 2018, 12, 11088, with similar device structure. Here, the CSS deposition of the Sb_2Se_3 absorber layer for photovoltaics application is not a novelty (Solar Energy Mater. Solar Cells, 2018, 188, 177, Sol. RRL, 2: 1800128.2018 and Nano Energy, 49, 346, 2018) although the authors claim the cooling rate could significantly impact on the photoelectrochemical performance.

Response #1:

Regarding the similarity to the previous works, using a similar device structure (ex, $RuO_x/TiO_2/Sb_2Se_3/Au/FTO$) does not necessarily mean the lack of novelty. For example, Pan et al, reported a benchmark performance of Cu_2O photocathodes ($RuO_x/TiO_2/Ga_2O_3/Cu_2O/Au$) and 3% STH efficiency by the Cu_2O photocathodes coupled with $BiVO_4$ photoanodes ($NiFeO_x/H,Mo:BiVO_4$) in *Nature Catalysis* (*Nat. Cat.*, **2018**, *1*,

412). Although all of the synthetic methods and device structures of both the photoanode and the photocathode had been already known, their tandem device result was worth to be published in the prestigious journal and has been cited more than 90 times up to the present. Moreover, our current work achieved lots of scientific advances in Sb₂Se₃ photocathodes in the following points of views;

- 1) The first demonstration of the cooling rate effect on the morphology of Sb₂Se₃
- 2) Insight toward synthesis meta-stable morphology during CSS deposition
- 3) Demonstration of the importance of pinhole-free films for high performance
- 4) The highest level of photocurrent among not only Sb₂Se₃ but also all photoelectrodes for PEC water splitting
- 5) The first unassisted water splitting by Sb₂Se₃-based photocathodes
- 6) A promising level of STH efficiency among photoanode-cathode tandem cells
- 7) The first demonstration of the role of V⁵⁺ ions in enhancing the stability of the PEC tandem devices (so far only tested in a half cell reaction)
- 8) Demonstration of the potential of Sb₂Se₃, as a low-cost p-type semiconductor which is one of the most important but less investigated field

We are not insisting that our result is the most impactful research in this decade. But we still believe that our result will significantly contribute to the PEC research field and is also worth to be published in *Nature Communications*.

Comment #2:

However, the fundamental mechanism for improved PEC performance is not clear, particularly, the chemical composition, microstructure did not show significant differences.

Response #2:

We should note that it is common sense in the PEC research field that direct contact between the top and bottom contact can cause significant degradation of the performance, even in the case of the chemical composition and optoelectronic properties have negligible differences. For example, Luo et al. reported the effect of a thin blocking layer to prevent shunt pathways in Cu₂O nanowire photocathodes (*Nano Lett.* **2016**, *16*, 1848). As shown in Fig. R1, the PEC performance of Cu₂O photocathodes significantly increased upon deposition of a very thin blocking Cu₂O layer. There were no significant differences in chemical composition, microstructure, and optoelectronic properties, thereby demonstrating the detrimental effect of

shunt pathways. Although the reviewer pointed out the fundamental mechanism for improved PEC performance is not clear, we strongly believe that we have rather proven the performance degradation mechanism in the presence of pinholes by showing rapid potential drops near pinholes by KPFM analysis, which had not yet been experimentally demonstrated. That is why the Reviewer #3 mentioned “Generating pinhole-free Sb_2Se_3 is crucial for long-term overall solar water splitting at zero bias.” In our revised manuscript, we have modified our manuscript to clarify the importance of preventing pin-holes formation in order to avoid any confusion of readers.

Fig. R1. The effect of a thin blocking layer preventing shunt pathways in PEC performance of Cu_2O photocathodes (*Nano Lett.* **2016**, *16*, 1848).

Revision made (colored blue):

(Line 18, Page 13)

In such a case, the photo-excited electrons can be extracted laterally to the ribbons and they can recombine with the holes at the back contact as shown in Fig. 3h due to the large electric field across the p-n junction. It is widely known that direct contact between the top and bottom contact can cause significant degradation of the performance, even in the case of the chemical composition and optoelectronic properties have negligible differences. For example, Luo et al. reported the effect of a thin blocking layer to prevent shunt pathways, thereby enabling much higher performance in Cu_2O nanowire photocathodes without any noticeable differences in chemical composition and morphology⁴³. The KPFM results clearly demonstrated the importance of pin-hole-free compact thin films in preventing the

recombination and the performance degradation mechanism in the presence of pin-holes, which had not yet been experimentally demonstrated. ~~as well as the origin of the performance difference between the fast and slow cooling samples.~~

43 J. Luo et al., Cu₂O Nanowire Photocathodes for Efficient and Durable Solar Water Splitting, *Nano Lett.*, **16**, 1848-1857 (2016).

Comment #3:

Particularly, the mechanism for the Au layer increase almost 3 times of the photocurrent (Fig.R4) has no detailed analysis, but only cite the previous work.

Response #3:

Regarding to the use of the Au layer, there are some comprehensive studies on hole-selective materials for Sb₂Se₃-based photocathodes and solar cells (*ACS Energy Lett.*, **2019**, *4*, 995; *Solar Energy*, **2019**, *182*, 96). In these papers, the authors have already reported the role of hole selective layers by comparing different types of hole selective layers as well as the performance enhancement mechanism while measuring the band positions and the resistivity, which had been cited in the present study. We believe that it is unnecessary to repeat all observations, which will make the present work deviated from the main point. Thus, it is absolutely valid to cite the previous well-known phenomena in our manuscript instead of repeating the details.

Comment #4:

And the low overpotential of the Sb₂Se₃, i.e., 0.4V still lower than other systems has not been investigated.

Response #4:

Here we respectfully disagree the reviewer's comment for the following reasons. First, our Sb₂Se₃ photocathode revealed the highest level of photocurrent and photovoltage among the previously reported Sb₂Se₃ photocathodes as shown in Fig. R2a. It should be also noted that the maximum photovoltage obtained by a semiconductor is in relation to its band gap. Fig. R2b shows the photovoltage benchmarks for PEC and PV materials as a function of optical band gap (modified from the figure in *Current Opinion in Electrochemistry*, **2017**, *2*, 104).

The Sb_2Se_3 photocathode in the present study is located at the black dashed line (the green star) which represents $\text{SQ}-0.4\text{ V}$. It is obvious that our Sb_2Se_3 photocathode achieved a higher figure of merit than most of the previous photoanodes and photocathodes, and actually revealed close to single crystalline photocathodes such as Si and InP. Thus, we strongly believe that our Sb_2Se_3 photocathodes revealed meaningfully high onset potential not only among previous Sb_2Se_3 photocathodes but also all other photoelectrode materials considering its small band gap.

Figure R2. (a) Previously reported photocurrent density and photovoltage obtained by some photocathode materials. ¹*J. Mater. Chem. A*, **2017**, *5*, 2180; ²*J. Mater. Chem. A*, **2017**, *5*, 23139; ³*ACS Nano*, **2017**, *11*, 12753; ⁴*Adv. Energy Mater.*, **2018**, *8*, 1702888; ⁵*ACS Appl. Mater. Interfaces*, **2018**, *10*, 10898; ⁶*Adv. Energy Mater.*, **2019**, *9*, 1900179; ⁷*ACS Energy Lett.*, **2019**, *4*, 517; ⁸*ACS Energy Lett.*, **2019**, *4*, 995 (b) Photovoltage benchmarks for PEC and PV materials as a function of optical band gap (*Current Opinion in Electrochemistry*, **2017**, *2*, 104)

Comment #5:

By considering the Pt or RuOx cocatalyst and the 70 nm Au hole collection layer are all precious materials in the RuOx/TiO2/Sb2Se3/Au/FTO and Pt/TiO2/CdS/Sb2Se/Au/FTO structure will significantly limit the scale application and may weaken the claimed "low cost" Sb2Se3 but an expensive device.

Response #5:

This comment is partially correct, but it is widely well-known that the most important building block in a PEC water splitting device in terms of cost-effectiveness is the light-

absorbing semiconductors. For example, the cost portion of electrocatalysts in the c-Si based PV-EC system is just approximately 1 % (whether the electrocatalyst is Ir-Ru or NiFe-NiMo) for overall cost of hydrogen production (*Energy Environ. Sci.*, **2014**, 7, 3828). Additionally, in the high STH systems based on III–V semiconductors, the cost of InGaP/GaAs (\$175) per unit solar collection area is much greater than other parts such as catalysts (Pt and IrO_x, \$8) and membranes (127 mm-thick Nafion, \$5) (*Energy Environ. Sci.*, **2016**, 9, 2354). Moreover, the community is recognizing that noble metals are easy to recycle, as well as price of hydrogen expected from non-noble metal electrocatalysts is not competitive to the one with noble metal ones (*Nat. Energy*, **2019**, 4, 430). It is also noteworthy that there is an alternative low-cost materials (Cu:NiO) for hole selective contact of Sb₂Se₃ photocathodes (*ACS Energy Lett.*, **2019**, 4, 995). That is also an important merit of Sb₂Se₃ photocathode for further reducing the overall cost, considering the fact that single crystalline semiconductors (such as GaAs and InP) cannot be grown on cost-effective substrates. Aforementioned descriptions are generally accepted in the PEC water splitting field, which is well supported by John Turner (who had been working in this field over 30 years) in his commentary article (*Science*, **2014**, 344, 469) in which the importance of the development of cost-effective semiconductors having good optoelectronic properties is highly emphasized. In the revised manuscript, we have added some descriptions to emphasize the importance of semiconducting materials in terms of cost-effectiveness of PEC water splitting devices.

Revision made (colored in blue):

(Line 11, Page 11)

Given the low-cost and relatively short history of Sb₂Se₃ as well as the simple preparation and low material usage due to the high α , the high photocurrent density of $\sim 30 \text{ mA cm}^{-2}$ at 0 V_{RHE} clearly demonstrates the strong potential of Sb₂Se₃ as a promising photocathode material. It is worth emphasizing that the most important building block in a PEC water splitting device in terms of cost-effectiveness is the light-absorbing semiconductors. For example, the cost portion of electrocatalysts in the c-Si based PV-EC system is just approximately 1 % (whether the electrocatalyst is Ir-Ru or NiFe-NiMo) for overall cost of hydrogen production³⁷. Additionally, in the high STH systems based on III–V semiconductors, the cost of InGaP/GaAs (\$175) per unit solar collection area is much greater than other parts such as catalysts (Pt and IrO_x, \$8) and membranes (127 mm-thick Nafion, \$5)³⁸. Moreover, the community is recognizing that noble metals are easy to recycle, as well

as price of hydrogen expected from non-noble metal electrocatalysts is not competitive to the one with noble metal ones³⁹. Thus, despite the use of relatively expensive catalysts and a hole selective contact layer, the high performance of our device demonstrates the feasibility of cost-effective Sb₂Se₃ based photoelectrodes for PEC water splitting.

(References)

37 C. A. Rodriguez et al., “Design and cost considerations for practical solar-hydrogen generators”, *Energy Environ. Sci.*, **7**, 3828-3835 (2014).

38 M. R. Shaner et al., “A comparative technoeconomic analysis of renewable hydrogen production using solar energy”, *Energy Environ. Sci.*, **9**, 2354-2371 (2016).

39 J. Kibsgaard & I. Chorkendorff, “Considerations for the scaling-up of water splitting catalysts”, *Nat. Energy.*, **4**, 430-433 (2019).

Comment #6:

By forming a tandem PEC cells with BiVO₄, the efficiency is not promising for 10 h stability. Overall, it does not show clearly innovative evidence and understanding to publish in Nature Communications when more similar results are published in other journals. Thus, I recommend the manuscript to be rejected and may submit to elsewhere.

Response #6:

As we presented in Fig 6. in the manuscript, 10 h stability is the 2nd longest record for PEC tandem devices (the first one is BiVO₄-Cu₂O tandem cells revealing 12 h stability). Our Sb₂Se₃ photocathode, operating stably over 30 h, is also the most stable Sb₂Se₃ photocathodes. Additionally, as we mentioned above, our work is the first demonstration of the role of V⁵⁺ ions in enhancing the stability of the PEC tandem devices, which is another novelty regarding the stability of our device. We are not insisting that our work represents a matured technology ready to commercialize, but we still believe that our work has sufficient novelty and significance to be published in *Nature Communications*.

Reviewer #2

Remarks to the Author:

The authors have carefully considered and thoroughly and satisfactorily responded to all of

my concerns, and the manuscript is now suitable for publication.

Response:

We are pleased to hear that the reviewer is satisfied with our response to the reviewer's comments. We thank the reviewer for his/hers effort for improving the overall quality of our manuscript.

Reviewer #3

Remarks to the Author

The authors have suitably addressed all of the criticisms by all reviewers. The authors are commended for their diligence in conducting new experiments and including new analysis to answer each question. This reviewer is convinced that the findings are sufficiently distinct and novel to warrant publication in *Nature Communications*. In particular, carefully probing the optoelectronic properties of the multi-junction device exposes relevant questions and limitations for each component, which adds to new materials chemistry. Generating pinhole-free Sb_2Se_3 is crucial for long-term overall solar water splitting at zero bias.

Response:

We thank the reviewer for evaluating our work as suitable to be published in *Nature Communications*. In particular, the remarks of the Reviewer #3, stating the importance of the pinhole-free Sb_2Se_3 as well as long-term overall solar water splitting, are appreciated because the reviewer seems to well understand the main point of our manuscript. We thank the reviewer again for their review works on improving the overall quality of our manuscript.